# CCIL: Continuity-based Data Augmentation for Corrective Imitation Learning

**Liyiming Ke**[*], **Yunchu Zhang**[*], **Abhay Deshpande, Siddhartha Srinivasa, Abhishek Gupta**
Paul G. Allen School of Computer Science and Engineering, University of Washington
`{kayke,yunchuz,abhayd,siddh,abhgupta}@cs.washington.edu`

## ABSTRACT

We present a new technique to enhance the robustness of imitation learning methods by generating corrective data to account for compounding errors and disturbances. While existing methods request interactive experts, additional offline datasets, or domain-specific invariances, our approach requires minimal additional assumptions beyond access to expert data. Our key insight is to leverage local continuity in the environment dynamics to generate corrective labels. Our method constructs a dynamics model from expert demonstrations, emphasizing local Lipschitz continuity in the learned model. In regions exhibiting local continuity, our algorithm generates corrective labels within the neighborhood of the demonstrations, extending beyond the actual set of states and actions in the dataset. Training on the augmented data improves the agent's resilience against perturbations and its capability to address compounding errors. To validate the efficacy of our generated labels, we conduct experiments across diverse robotics domains in simulation, encompassing classic control problems, drone flying, navigation with high-dimensional sensor observations, legged locomotion, and tabletop manipulation.

## 1 INTRODUCTION

The application of imitation learning in real-world robotics necessitates extensive data, and the success of simple yet practical methods, such as behavior cloning (Pomerleau, 1988), relies on datasets with comprehensive coverage (Bojarski et al., 2016; Florence et al., 2022). However, obtaining such datasets can be prohibitively expensive, particularly in domains like robots that lacks a readily available plethora of expert demonstrations. Furthermore, robotic policies can behave unpredictably and dangerously when encountering states not covered in the expert dataset, due to various factors such as sensor noise, stochastic environments, external disturbances, or compounding errors leading to covariate shift (Ross et al., 2011; de Haan et al., 2019). For widespread deployment of imitation learning in real-world robotic applications, policies need to ensure their robustness even when encountering unfamiliar states.

Many strategies to address this challenge revolve around augmenting the training dataset. Classic techniques for data augmentation require either an interactive expert (Ross et al., 2011; Laskey et al., 2017) or knowledge about the systems' invariances (Bojarski et al., 2016; Florence et al., 2019; Zhou et al., 2023). This information is necessary to inform the learning agent how to recover from out-of-distribution states. However, demanding such information can be burdensome (Ke et al., 2021b), expensive (Zhou et al., 2023), or practically infeasible across diverse domains. As a result, behavior cloning, which assumes only access to expert demonstrations, has remained the de-facto solution in many domains (Silver et al., 2008; Rahmatizadeh et al., 2018; Ke et al., 2021b; Florence et al., 2022; Chen et al., 2022; Zhou et al., 2022).

For general applicability, this work focuses on imitation learning (IL) that rely *solely on expert demonstrations*. We propose a method to enhance robustness of IL by generating corrective labels for data augmentation. We recognize an under-exploited feature of dynamic systems: local continuity. Despite the complex transitions and representations in system dynamics, they need to adhere to the laws of physics and exhibit some level of local continuity, where small changes to actions or states result in small changes in transitions. While realistic systems may contain discontinuities in certain state space portions, the subset exhibiting local continuity proves to be a valuable asset.

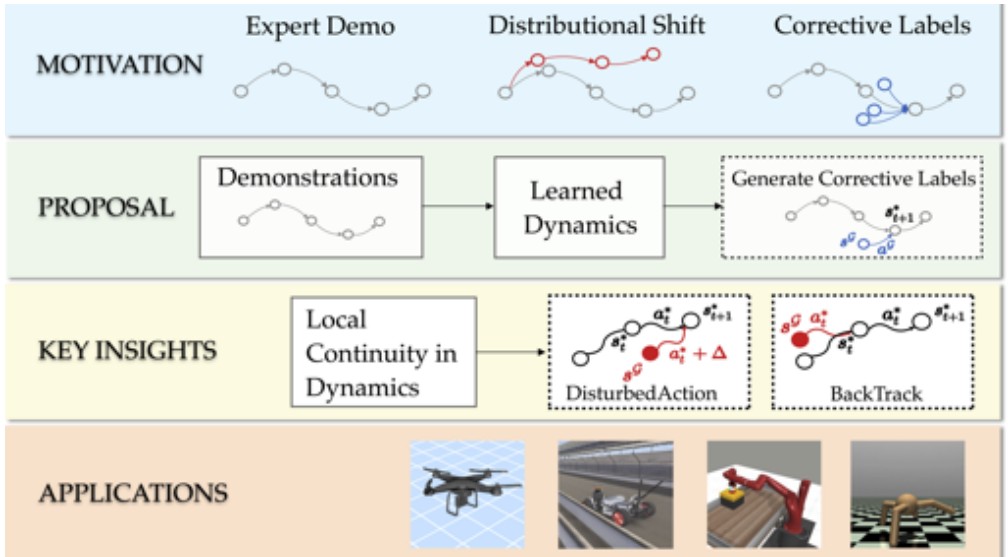

Figure 1: **Our proposed framework, CCIL.** To enhance the robustness of imitation learning, we propose to augment the dataset with synthetic corrective labels. We leverage the local continuity in the dynamics, learn a regularized dynamics function and generate corrective labels *near* the expert data support. We provide theoretical guarantees on the quality of the generated labels. We present empirical evaluations CCIL over 14 robotic tasks in 4 domains to showcase CCIL improving imitation learning agents' robustness to disturbances.

Armed with this insight, our goal is to synthesize *corrective* labels that guide an agent encountering unfamiliar states back to the distribution of expert states. Leveraging the presence of local continuity makes the synthesis of these corrective labels more tractable. A learned dynamics model with local Lipschitz continuity can navigate an agent from unfamiliar out-of-distribution states to in-distribution expert trajectories, even extending beyond the expert data support. The local Lipschitz continuity allows the model to have bounded error in regions *outside* the expert data support, providing the ability to generate appropriately corrective labels.

We propose a mechanism for learning dynamics models with local continuity from expert data and generating corrective labels. Our practical algorithm, **CCIL**, leverages local **C**ontinuity in dynamics to generate **C**orrective labels for **I**mitation **L**earning. We validate CCIL empirically on a variety of robotic domains in simulation. In summary, our contributions are:

- **Problem Formulation:** We present a formal definition of *corrective labels* to enhance robustness for imitation learning. (Sec. 4.1).
- **Practical Algorithm:** We introduce **CCIL**, a framework for learning dynamics functions, and leveraging local continuity to generate corrective labels. Our method is lean on assumptions, primarily relying on availability of expert demonstrations and the presence of local continuity in the dynamics.
- **Theoretical Guarantees:** We showcase how local continuity in dynamic systems would allow extending a learned model's capabilities beyond the expert data. We present practical methods to enforce desired local smoothness while fitting a dynamics function to the data and accommodating discontinuity (Sec. 4.2). We provide a theoretical bound on the quality of the model in this area and the generated labels (Sec. 4.3).
- **Extensive Empirical Validation:** We conduct experiments over 4 distinct robotic domains across 14 tasks in simulation, ranging from classic control, drone flying, high-dimensional car navigation, legged locomotion and tabletop manipulation. We showcase our proposal's ability to enhance the performance and robustness of imitation learning agents (Sec. 5).

## 2 RELATED WORK

**Imitation Learning (IL) and Data Augmentation.** Given only the expert demonstrations, Behavior Cloning (BC) remains a strong empirical baseline for imitation learning (Pomerleau, 1988). It formulates IL as a supervised learning problem and has a plethora of data augmentation methods. However,

previous augmentation methods mostly leverage expert or some form of invariance (Venkatraman et al., 2015; Bojarski et al., 2016; Florence et al., 2019; Spencer et al., 2021; Zhou et al., 2023), which is a domain-specific property. Ke et al. (2021b) explores noise-based data augmentation but lacks principle for choosing appropriate noise parameters. Park & Wong (2022) learns dynamics model for data augmentation, similar to our approach. But it learns an *inverse* dynamics model and lacked theoretical insights. In contrast, our proposal leverages local continuity in the dynamics, is agnostic to domain knowledge, and provides theoretical guarantees on the quality of generated data.

**Mitigating Covariate Shift in Imitation Learning.** Compounding errors push the agent astray from expert demonstrations. Prior works addressing the covariate shift often request additional information. Methods like DAGGER (Ross et al., 2011), LOLS (Chang et al., 2015), DART (Laskey et al., 2017) and AggrevateD (Sun et al., 2017) use interactive experts, while GAIL (Ho & Ermon, 2016), SQIL (Reddy et al., 2019) and AIRL (Fu et al., 2017) sample more transitions in the environment to minimize the divergence of states distribution between the learner and expert (Ke et al., 2021a; Swamy et al., 2021). Offline Reinforcement Learning methods like MOREL (Kidambi et al., 2020) are optimized to mitigate covariate shift but demand access to a ground truth reward function. Reichlin et al. (2022) trains a recovery policy to move the agent back to data manifold by assuming that the dynamics is known. MILO (Chang et al., 2021) learns a dynamics function to mitigate covariate shift but requires access to abundant offline data to learn a high-fidelity dynamics function. In contrast, our proposal is designed for imitation learning without requiring additional data or feedback, complementing existing IL methods. We include MILO and MOREL as baselines in experiments.

**Local Lipschitz Continuity in Dynamics.** Classical control methods often assume local Lipschitz continuity in the dynamics to guarantee the existence and uniqueness of solutions to differential equations. For example, the widely adopted $\mathcal{C}^2$ assumption in optimal control theory (Bonnard et al., 2007) and the popular control framework iLQR (Li & Todorov, 2004). This assumption is particularly useful in the context of nonlinear systems and is prevalent in modern robot applications (Seto et al., 1994; Kahveci, 2007; Sarangapani, 2018). However, most of these methods leveraging dynamics continuity are in the optimal control setting, requiring a pre-specified dynamics model and cost function. In contrast, this work focuses on robustifying imitation learning agents by learning a locally continuous dynamics model from data, without requiring a human-specified model or cost function.

**Learning Dynamics using Neural Networks.** Fitting a dynamics function from data is an active area of research (Hafner et al., 2019; Wang et al., 2022; Kaufmann et al., 2023). For continuous states and transitions, Asadi et al. (2018) proved that enforcing Lipschitz continuity in training dynamics model could lower the multi-step prediction errors. Ensuring *local* continuity in the trained dynamics, however, can be challenging. Previous works enforced Lipschitz continuity in training neural networks (Bartlett et al., 2017; Arjovsky et al., 2017; Miyato et al., 2018), but did not apply it to dynamics modeling or generating corrective labels. Shi et al. (2019) learned smooth dynamics functions via enforcing *global* Lipschitz bounds and demonstrates on the problem of drone landing. Pfrommer et al. (2021) learned a smooth model for fictional contacts for manipulation. While this work is not focusing on learning from visual inputs, works such as Khetarpal et al. (2020); Zhang et al. (2020); Zhu et al. (2023) are actively building techniques for learning dynamics models from high-dimensional inputs that could be leveraged in conjunction with the insights from our proposal.

**Concurrent Works** In the final stages of preparing our manuscript, we became aware of a concurrent work (Block et al., 2024) that shares similar motivations with our study - to stabilize imitation learning near the expert demonstrations. Instead of data augmentation, their proposal directly generates stabilizing controller. We hope both works can serve as a foundation for future research into robustifying imitation learning.

## 3 PRELIMINARIES

We consider an episodic finite-horizon Markov Decision Process (MDP), $\mathcal{M} = \{\mathcal{S}, \mathcal{A}, f, P_0\}$, where $\mathcal{S}$ is the state space, $\mathcal{A}$ is the action space, $f$ is the ground truth dynamics function and $P_0$ is the initial state distribution. A *policy* maps a state to a distribution of actions $\pi \in \Pi : s \to a$. The dynamics function $f$ can be written as a m apping from a state $s_t$ and an action $a_t$ at time $t$ to the change to a new state $s_{t+1}$. This is often represented in its residual form: $s_{t+1} = s_t + f(s_t, a_t)$. Following the imitation learning setting, the true dynamics function $f$ is unknown, the reward is unknown, and we only have transitions drawn from the system dynamics.

We are given a set of demonstrations $\mathcal{D}^*$ as a collection of transition triplets: $\mathcal{D}^* = \{(s_j^*, a_j^*, s_{j+1}^*)\}_j$, where $s_j^*, s_{j+1}^* \in \mathcal{S}, a_j^* \in \mathcal{A}$. We can learn a policy from these traces via supervised learning (behavior cloning) by maximizing the likelihood of the expert actions being produced at the training states: $\arg\max_{\hat{\pi}} \mathbb{E}_{s_j^*, a_j^*, s_{j+1}^* \sim \mathcal{D}^*} \log(\hat{\pi}(a_j^* \mid s_j^*))$. In practice, we can optimize this objective by minimizing the mean-squared error regression using standard stochastic optimization.

## 4 GENERATING CORRECTIVE LABELS FOR ROBUST IMITATION LEARNING LEVERAGING CONTINUITY IN DYNAMICS

Our objective is to enhance the robustness of imitation learning methods by generating corrective labels that bring an agent from out-of-distribution states back to in-distribution states. We first define the desired *high quality corrective* labels to make imitation learning more robust in Sec. 4.1. We then discuss how these labels can be generated with a known ground truth dynamics of the system (App. A). Without the ground truth dynamics, we discuss how this can be done using a *learned* dynamics model. Importantly, to generate high-quality corrective labels using learned dynamics functions, the dynamics need to exhibit local continuity. We present a method to train a locally Lipschitz-bounded dynamics model in Sec. 4.2, and then show how to use such a learned model to generate corrective labels (Sec. 4.3) with high confidence, beyond the support of the expert training data. Finally, in Section 4.4, we instantiate these insights into a practical algorithm - **CCIL** that improves the robustness of imitation learning methods with function approximation.

### 4.1 CORRECTIVE LABELS FORMULATION

To robustify imitation learning, we aim to provide corrections to disturbances or compounding errors by bringing the agent back to the "known" expert data distribution, where the policy is likely to be successful. We generate state-action-state triplets $(s^{\mathcal{G}}, a^{\mathcal{G}}, s^*)$ that are *corrective*: executing action $a^{\mathcal{G}}$ in state $s^{\mathcal{G}}$ can bring the agent back to a state $s^*$ in support of the expert data distribution.

**[Corrective Labels].** $(s^{\mathcal{G}}, a^{\mathcal{G}}, s^*)$ is a corrective label triplet with starting state $s^{\mathcal{G}}$, corrective action label $a^{\mathcal{G}}$ and target state (in the expert data) $s^* \in \mathcal{D}^*$ if, with respect to dynamics function $f$,

$$\|[s^{\mathcal{G}} + f(s^{\mathcal{G}}, a^{\mathcal{G}})] - s^*\| \leq \epsilon_c. \tag{1}$$

This definition is trivially satisfied by the given expert demonstration $(s_j^*, a_j^*, s_{j+1}^*) \in \mathcal{D}^*$. We aim to search for a larger set of corrective labels for out-of-distribution states. If the policy has bounded error on not just the distribution of expert states but on a distribution with larger support, $\text{Support}(d^{\pi^*}) < \text{Support}(d^{\pi_{\text{aug}}})$, it is robust against a larger set of states that it might encounter during execution. However, without knowledge of the true system dynamics $f$, we have only an approximation $\hat{f}$ of the dynamics function derived from finite samples.

**Definition 4.1. [High Quality Corrective Labels under Approximate Dynamic Models].** $(s^{\mathcal{G}}, a^{\mathcal{G}}, s^*)$ is a corrective label if $\|[s^{\mathcal{G}} + \hat{f}(s^{\mathcal{G}}, a^{\mathcal{G}})] - s^*\| \leq \epsilon_c$, w.r.t. an approximate dynamics $\hat{f}$ for a target state $s^* \in \mathcal{D}^*$. Such a label is "high-quality" if the approximate dynamics function also has bounded error w.r.t. the ground truth dynamics function evaluated at the given state-action pair $\|f(s^{\mathcal{G}}, a^{\mathcal{G}}) - \hat{f}(s^{\mathcal{G}}, a^{\mathcal{G}})\| \leq \epsilon_{co}$.

The high-quality corrective labels represent the learned dynamics model's best guess at bringing the agent back into the support of expert data. However, an approximate dynamics model is only reliable in a certain region of the state space, i.e., where the predictions of $\hat{f}$ closely align with the true dynamics of the system, $f$. When the dynamics model is trained without constraints, such as in a maximum likelihood framework, its performance might significantly degrade when extrapolating beyond the distribution of the training data. To address this challenge, we will first describe how to learn locally continuous dynamics models from data. Subsequently, we will outline how to utilize these models for high-quality corrective label generation beyond the expert data.

### 4.2 LEARNING LOCALLY CONTINUOUS DYNAMICS MODELS FROM DATA

Our core insight for learning dynamics models and using them beyond the training data is to leverage the local continuity in dynamic systems. When the dynamics are locally Lipschitz bounded, small

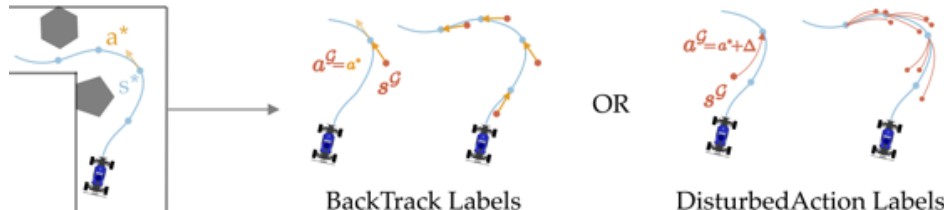

BackTrack Labels           DisturbedAction Labels

Figure 2: **Generating corrective labels from demonstration.** Given expert demonstration state $s^*$ and action $a^*$ that arrives at $s^*_{next}$. BackTrack Labels: search for an alternative starting state $s^{\mathcal{G}}$ that can arrive at expert state $s^*$ if it executes expert action $a^*$. DisturbedAction Labels: search for an alternative starting state $s^{\mathcal{G}}$ that would arrive at $s^*_{next}$ if it executes a slightly disturbed expert action $a^{\mathcal{G}} = a^*_t + \Delta$, where $\Delta$ is a sampled noise.

changes in state and/or action yield small changes in the transitions. A dynamics function that exhibit locally continuity would allow us to extrapolate beyond the training data to the states and actions that are in proximity to the expert demonstration - a region in which we can trust the learned model.

We follow the model-based reinforcement learning framework to learn dynamics models from data (Wang et al., 2019). Essentially, we learn a residual dynamics model $\hat{f}(s^*_t, a^*_t) \to s^*_{t+1} - s^*_t$ via mean-squared error regression (MSE). We can use any function approximator (e.g., neural network) and write down the learning loss:

$$\arg \min_{\hat{f}} \mathbb{E}_{s^*_j, a^*_j, s^*_{j+1} \sim \mathcal{D}^*} [\text{MSE}], \quad \text{MSE} = \|\hat{f}(s^*_j, a^*_j) + s^*_j - s^*_{j+1}\|. \tag{2}$$

A model trained with the MSE loss alone is not guaranteed to have good extrapolation beyond the training data. Critically, we modify the above learning objective to ensure the learned dynamics model to contain regions that are locally continuous. Previously Shi et al. (2019) used spectral normalization to fit dynamics functions that are *globally* Lipschitz bounded. However, most robotics problems involve dynamics models that are hybrids of local continuity and discontinuity: for instance, making and breaking contacts. In face of discontinuity, we present a few practical methods to fit a dynamics model that (1) enforces as much local continuity as permitted by the data and (2) discards the discontinuous regions when subsequently generating corrective labels. Due to space limit, we defer the reader to App. C.1 for a complete list of candidate loss functions considered. Here, we show an example of sampling-based penalty of violation of local Lipschitz constraint.

**Local Lipschitz Continuity via Sampling-based Penalty**. Following Gulrajani et al. (2017), we relax the global Lipschitz constraint by penalizing any violation of the local Lipschitz constraint.

$$\arg \min_{\hat{f}} E_{s^*_j, a^*_j, s^*_{j+1} \sim \mathcal{D}^*} \left[ \text{MSE} + \lambda \cdot \mathbb{1} \big( \|\hat{f}'(s^*_j, a^*_j)\| > L \big) \right] \tag{3}$$

$$\hat{f}'(s^*_j, a^*_j) \approx \mathbb{E}_{\Delta_s \sim \mathcal{N}} \Big[ \frac{\hat{f}(s^*_j + \Delta_s, a^*_j) - \hat{f}(s^*_j, a^*_j)}{\Delta_s} \Big] \tag{4}$$

We use samples to estimate the local Lipschtiz continuity $\hat{f}'(s^*_j, a^*_j)$, by perturbing the data points with small sampled Gaussian noises $\Delta_s \sim \mathcal{N}$. Alternatively, one can compute the Jacobian matrix to estimate the local Lipschitz continuity. The penalty term weighted by $\lambda$ enforces the local Lipschitz constraint at the specific expert datapoint. Doing so ensures that the approximate model is mostly $L$ Lipschitz-bounded while being predictive of the transitions in the expert data. This approximate dynamics model can then be used to generate corrective labels.

### 4.3 GENERATING HIGH-QUALITY CORRECTIVE LABELS

We employ learned dynamics models to generate corrective labels, leveraging local continuity to ensure the generated labels have bounded error in proximity to the expert data's support. When the local dynamics function is bounded by a Lipschitz constant w.r.t. the state-action space, we can (1) perturb the dynamics function by introducing slight variations in either state or action and (2)

quantify the prediction error from the dynamics model based on the Lipschitz constant. We outline two techniques, BackTrack label and DisturbedAction label, to generate corrective labels (Fig. 2).

**Technique 1: BackTrack Label.** Assuming local Lipschitz continuity w.r.t. states, given expert state-action pair $s_t^*, a_t^*$, we propose to find a different state $s_{t-1}^{\mathcal{G}}$ that can arrive at $s_t^*$ using action $a_t^*$. To do so, we optimize for a state $s_{t-1}^{\mathcal{G}}$ such that:

$$s_{t-1}^{\mathcal{G}} + \hat{f}(s_{t-1}^{\mathcal{G}}, a_t^*) = s_t^* \tag{5}$$

The quality of the generated label $(s^{\mathcal{G}}, a_t^*, s_t^*)$ is bounded and we present the proof in Appendix B.1.

**Theorem 4.2.** *When the dynamics model has a training error of $\epsilon$ on the specific data point, under the assumption that the dynamics models $f$ and $\hat{f}$ are locally Lipschitz continuous w.r.t. state with Lipschitz constant $K_1$ and $K_2$ respectively, if $s_t^{\mathcal{G}}$ is in the neighborhood of local continuity, then*

$$\left\| f\left(s_t^{\mathcal{G}}, a_{t+1}^*\right) - \hat{f}\left(s_t^{\mathcal{G}}, a_{t+1}^*\right) \right\| \leq \epsilon + (K_1 + K_2) \left\| s_t^{\mathcal{G}} - s_{t+1}^* \right\|. \tag{6}$$

**Technique 2: DisturbedAction.** Assuming local Lipschitz continuity w.r.t., state-action, given an expert tuple $(s_t^*, a_t^*, s_{t+1}^*)$, we ask: Is there an alternative action $a^{\mathcal{G}}$ that slightly differs from the demonstrated action $a_t^*$, i.e., $a^{\mathcal{G}} = a_t^* + \Delta$, that can bring an imaginary state $s^{\mathcal{G}}$ to the expert state $s_{t+1}^*$? We randomly sample a small action noise $\Delta$ and solve for an imaginary previous state $s_t^{\mathcal{G}}$:

$$s_t^{\mathcal{G}} + \hat{f}(s_t^{\mathcal{G}}, a_t^* + \Delta) - s_{t+1}^* = 0. \tag{7}$$

For every data point, we can obtain a set of labels $(s_t^{\mathcal{G}}, a_t^* + \Delta)$ by randomly sampling the noise vector $\Delta$. We show that the quality of the labels is bounded (proof in App. B.2).

**Theorem 4.3.** *Given $s_{t+1}^* - \hat{f}(s_t^{\mathcal{G}}, a_t^* + \Delta) - s_t^{\mathcal{G}} = \epsilon$ and that the dynamics function $f$ is locally Lipschitz continuous w.r.t. states and actions, with Lipschitz constants $K_A$ and $K_S$ respectively, then*

$$\|f(s_t^{\mathcal{G}}, a_t^* + \Delta) - \hat{f}(s_t^{\mathcal{G}}, a_t^* + \Delta)\| \leq K_A \|\Delta\| + (1 + K_S)\|\epsilon\|. \tag{8}$$

**Solving the Root-Finding Equation in Generating Labels** Both our techniques need to solve root-finding equations (Eq. 5 and 7). This suggests an optimization problem: $\arg\min_{s^{\mathcal{G}}} \|s^{\mathcal{G}} + \hat{f}(s^{\mathcal{G}}, a^{\mathcal{G}}) - s^*\|$ given $\hat{f}$ and $a^{\mathcal{G}}, s^*$. There are multiple ways to solve this optimization problem (e.g., gradient descent or Newton's method). For simplicity, we choose a fast-to-compute and conceptually simple solver, Backward Euler, widely adopted in modern simulators (Todorov et al., 2012). It lets us use the gradient of the next state to recover the earlier state. To generate data given $a^{\mathcal{G}}, s^*$, Backward Euler solves a surrogate equation iteratively: $s^{\mathcal{G}} \leftarrow s^* - \hat{f}(s^*, a^{\mathcal{G}})$.

**Rejection Sample.** We can reject the generated labels if the root solver returns a solution that result in a large error bound, i.e., $\|s_t^* - s_t^{\mathcal{G}}\| > \epsilon$, where $\epsilon$ is a hyper-parameter. Rejecting labels that are outside a chosen region lets us directly control the size of the resulting error bound.

## 4.4 CCIL: CONTINUITY-BASED CORRECTIVE LABELS FOR IMITATION LEARNING

We instantiate a practical version of our proposal, **CCIL** (**C**ontinuity-based data augmentation to generate **C**orrective labels for **I**mitation **L**earning), as shown in Alg 1, with three steps: (1) fit an approximate dynamics function that exhibit local continuity, (2) generate corrective labels using the learned dynamics, and (3) use rejection sampling to select labels with the desired error bounds. We highlight the difference in our two techniques in color. To use the generated data $D^{\mathcal{G}}$, we simply combine it with the expert data and train policies using standard behavior cloning on the augmented dataset. We evaluate the augmented agent to empirically test whether training with synthetic corrective labels would help imitation learning.

## 5 EXPERIMENTS

We evaluate CCIL over 4 simulated robotic domains of 14 tasks to answer the following questions:
**Q1**. Can we empirically verify the theoretical contributions on the quality of the generated labels;

**Algorithm 1** CCIL: **C**ontinuity-based data augmentation for **C**orrective labels for **I**mitation **L**earning

1: **Input:** $\mathcal{D}* = (s_i^*, a_i^*, s_{i+1}^*)$
2: **Initialize:** $D^{\mathcal{G}} \leftarrow \varnothing$
3: LearnDynamics $\hat{f}$
4: **for** $i = 1..n$ **do**
5: $\quad (s_i^{\mathcal{G}}, a_i^{\mathcal{G}}) \leftarrow$ GenLabels $(s_i^*, a_i^*, s_{i+1}^*)$
6: $\quad$ **if** $||s_i^{\mathcal{G}} - s_i^*|| < \epsilon$ **then**
7: $\quad\quad \mathcal{D}^{\mathcal{G}} \leftarrow \mathcal{D}^{\mathcal{G}} \cup (s_i^{\mathcal{G}}, a_i^{\mathcal{G}})$
8: $\quad$ **end if**
9: **end for**
10: **Function** LearnDynamics $\hat{f}$
11: $\quad$ Optimize a chosen objective from Sec. 4.2
12: **Function** GenLabels(BackTrack)
13: $\quad a_i^{\mathcal{G}} \leftarrow a_i^*$
14: $\quad s_i^{\mathcal{G}} \leftarrow \arg\min_{s_i^{\mathcal{G}}} ||s_i^{\mathcal{G}} + \hat{f}(s_i^{\mathcal{G}}, a_i^{\mathcal{G}}) - s_i^*||$
15: **Function** GenLabels(DisturbedAction)
16: $\quad a_i^{\mathcal{G}} \leftarrow a_i^* + \Delta, \Delta \sim \mathcal{N}(0, \Sigma)$
17: $\quad s_i^{\mathcal{G}} \leftarrow \arg\min_{s_i^{\mathcal{G}}} ||s_i^{\mathcal{G}} + \hat{f}(s_i^{\mathcal{G}}, a^{\mathcal{G}}) - s_{i+1}^*||$

**Q2**. How well does CCIL handle discontinuities in environmental dynamics;
**Q3**. How would training with CCIL-generated labels affect imitation learning agents' performance.

We compare CCIL to standard Behavior Cloning (BC) (Pomerleau, 1988), NoisyBC (Ke et al., 2021b), MILO (Chang et al., 2021) and MOREL (Kidambi et al., 2020) using the same model architecture across all methods, over 10 random seeds. Note that MOREL requests additional access to reward function. Appendix. D contains details to reproduce the experiments.

## 5.1 ANALYSIS ON CLASSIC CONTROL OF PENDULUM AND DISCONTINUOUS PENDULUM

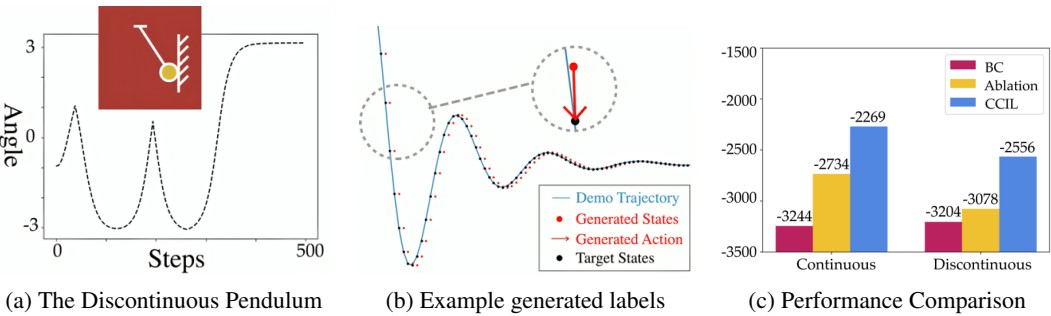

(a) The Discontinuous Pendulum  (b) Example generated labels  (c) Performance Comparison

Figure 3: Evaluation on the Pendulum and Discontinuous Pendulum Task.

We consider the classic control task, the pendulum, and also consider a variant, The Discontinuous Pendulum, by bouncing the ball back at a few chosen angles, as shown in Fig. 3a.

**Verifying the quality of the generated labels (Q1)**. We visualize a subset of generated labels in Fig. 3b. Note that the generated states (red) are outside the expert support (blue) and that the generated actions are torque control signals, which are challenging for invariance-based data augmentation methods to generate. To quantify the quality of the generated labels, we use the fact that ground truth dynamics can be analytically computed for this domain and measure how closely our labels can bring the agent to the expert. We observed an average L2 norm distance of 0.02367 which validates our derived theoretical bound of 0.065: $K1|\delta| + (1 + K2)|\epsilon| = 12 * 0.0001 + 13 * 0.005$ (Equation 6).

**The Impact of Local Lipschitz Continuity Assumption (Q2)** To highlight how local discontinuity in the dynamics affects CCIL, we compare CCIL performance in the continuous and discontinuous pendulum in Fig. 3c: CCIL improved behavior cloning performance even when discontinuity is present, albeit with a smaller margin for the discontinuous Pendulum. In an ablation study, we also tried generating labels using a naive dynamics model (without explicitly assuming Lipschitz continuity) which performed slightly better than vanilla behavior cloning but worse than CCIL, highlighting the importance of enforcing local Lipschitz continuity in training dynamics functions for generating corrective labels.

**CCIL improved behavior cloning agent (Q3)**. For both the continuous and discontinuous Pendulum, CCIL outperformed behavior cloning given the same amount of data (Fig. 3c).

## 5.2 Driving with Lidar: High-dimensional state input

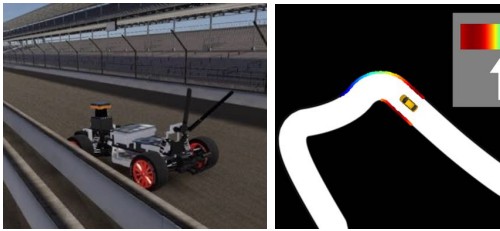

| Method | Succ. Rate | Avg. Score |
|--------|-----------|-----------|
| Expert | 100.0% | 1.00 |
| Morel | 0.0% | $0.001 \pm 0.001$ |
| MILO | 0.0% | $0.21 \pm 0.003$ |
| BC | 31.9% | $0.58 \pm 0.25$ |
| NoiseBC | 39.3% | $0.62 \pm 0.28$ |
| **CCIL** | **56.4%** | $\mathbf{0.75} \pm 0.25$ |

Figure 4: F1tenth  Figure 5: LiDar POV  Table 1: Performance on Racing

We compare all agents on the F1tenth racing car simulator (Fig. 4), which employs a high-dimensional, LiDAR sensor as input (Fig. 5). We evaluate each agent over 100 trajectories. At the beginning of each trajectory, the car is placed at a random location on the track. It needs to use the LiDAR input to decide on speed and steering, earning scores for driving faster or failing by crashing.

**CCIL can improve the performance of behavior cloning for high-dimensional state inputs (Q2, Q3)**. Table. 1 shows that CCIL demonstrated an empirical advantage over all other agents, achieving a higher success rate and better score while having fewer crashes. Noticeably, LiDar inputs can contain highly-complicated discontinuities and impose challenges to model-based methods (Morel and MILO), while CCIL could reliably generate corrective labels with confidence.

## 5.3 Drone Navigation: High-Frequency Control Task and Sensitive to Noises

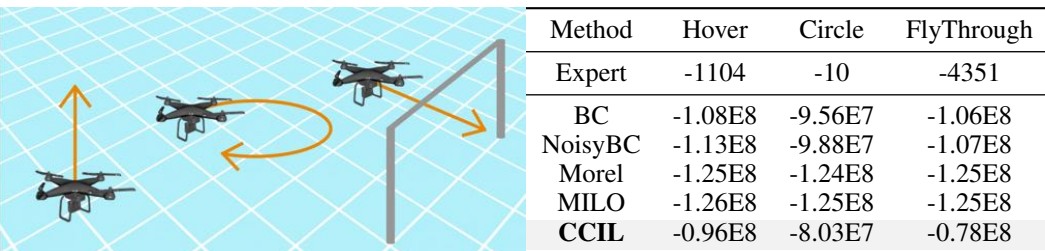

| Method | Hover | Circle | FlyThrough |
|--------|-------|--------|-----------|
| Expert | -1104 | -10 | -4351 |
| BC | -1.08E8 | -9.56E7 | -1.06E8 |
| NoisyBC | -1.13E8 | -9.88E7 | -1.07E8 |
| Morel | -1.25E8 | -1.24E8 | -1.25E8 |
| MILO | -1.26E8 | -1.25E8 | -1.25E8 |
| **CCIL** | -0.96E8 | -8.03E7 | -0.78E8 |

Figure 6: Drone Flying Tasks  Table 2: Performance on Drone

Drone navigation is a high-frequency control task and can be very sensitive to noise, making it an appropriate testbed for the robustness of imitation learning. We use an open-source quadcopter simulator, gym-pybullet-drone (Panerati et al., 2021) and consider three proposed tasks: hover, circle, fly-through-gate, as shown in Fig. 6. Following Shi et al. (2019), we inject observation and action noises during evaluation to highlight the robustness of the learner agent.

**CCIL improved performance for imitation learning agent and its robustness to noises (Q3)**. CCIL outperformed all baselines by a large margin.

## 5.4 Locomotion and Manipulation: Diverse Scenes with Varying Discontinuity

Manipulation and locomotion tasks commonly involve complex forms of contacts, raising considerable challenges for learning dynamics models and for our proposal. We evaluate the applicability of CCIL in such domains, considering 4 tasks from MuJoCo locomotion suite: Hopper, Walker2D, Ant, HalfCheetah and 4 tasks from the MetaWorld manipulation suites: CoffeePull, ButtonPress, CoffeePush, DrawerClose. During evaluation, we add a small amount of randomly sampled Gaussian noise to the sensor (observation state) and the actuator (action) to simulate real-world conditions of robotics controllers and to test the robustness of the agents.

**CCIL outperforms behavior cloning or at least achieves comparable performance even when varying form of discontinuity is present (Q2, Q3)**. On 4 out of 8 tasks considered, CCIL outperforms all baselines. Across all tasks, CCIL at least achieves comparable results to vanilla behavior cloning, shown in Table. 3, indicating that it's a effective alternative to behavior cloning without necessitating substantial extra assumptions.

Table 3: Evaluation results for Mujoco and Metaworld tasks with noise disturbances. We list the expert scores in a noise-free setting for reference. In the face of varying discontinuity from contacts, CCIL remains the leading agent on 4 out of 8 tasks (Hopper, Walker, HalfCheetah, CoffeePull). Comparing CCIL with BC: across all tasks, CCIL can outperform vanilla behavior cloning or at least achieve comparable performance.

| | Mujoco | | | | Metaworld | | | |
|---|---|---|---|---|---|---|---|---|
| | Hopper | Walker | Ant | Halfcheetah | CoffeePull | ButtonPress | CoffeePush | DrawerClose |
| Expert | 3234.30 | 4592.30 | 3879.70 | 12135.00 | 4409.95 | 3895.82 | 4488.29 | 4329.34 |
| BC | 1983.98 ± 672.66 | 1922.55 ± 1410.09 | 2965.20 ± 202.71 | 1798.98 ± 791.89 | 3552.59 ±233.41 | **3693.02** ± 104.99 | 1288.19± 746.37 | 3247.06 ± 468.73 |
| Morel | 152.19±34.12 | 70.27 ± 3.59 | 1000.77 ± 15.21 | -2.24 ± 0.02 | 18.78±0.09 | 14.85±17.08 | 18.66± 0.02 | 1222.23± 1241.47 |
| MILO | 566.98±100.32 | 526.72±127.99 | 1006.53±160.43 | 151.08±117.06 | 232.49± 110.44 | 986.46±105.79 | 230.62±19.37 | **4621.11**±39.68 |
| NoiseBC | 1563.56 ± 1012.02 | 2893.21 ± 1076.89 | **3776.65** ± 442.13 | 2044.24 ±291.59 | 3072.86 ± 785.91 | 3663.44±63.10 | **2551.11**± 857.79 | 4226.71± 18.90 |
| **CCIL** | **2631.25** ± 303.86 | **3538.48** ± 573.23 | 3338.35 ± 474.17 | **8893.81** ± 472.70 | **4168.46** ± 192.98 | 3775.22±91.24 | 2484.19± 976.03 | 4145.45± 76.23 |

## 5.5 SUMMARY

We conducted extensive experiments to address three research queries. For **Q1**, we verified the proposed theoretical bound on the quality of the generated labels on the Pendulum task. For **Q2**, we tested that CCIL is robust to environmental discontinuities, improving behavior cloning in both continuous (Pendulum, Drone) and discontinuous scenarios (Driving, MuJoCo). For **Q3**, CCIL emerged as the best agent on 10 out of 14 tasks. In the rest of tasks, it achieved comparable performance to behavior cloning, demonstrating its cost-effectiveness and efficiency as an alternative approach without requiring significant additional assumptions.

## 6 CONCLUSION

We address the problem of encouraging imitation learning agents to adhere closely to demonstration data support. We introduce corrective labels and propose **CCIL**, a novel framework that leverages local continuity in environmental dynamics to generate corrective labels. The key insight is to use the expert data to train a dynamics model that exhibits local continuity while skipping the discontinuous regions. This model, in turn, can augment imitation learning with corrective labels. While the assumption is that system dynamics exhibiting some local continuity, we've validated the effectiveness of our method on various simulated robotics environments, including drones, driving, locomotion and manipulation.

**Limitation: Optimality of Corrective Labels** While corrective labels constrain the learner distribution within the expert distribution, similar to a stabilizing controller, they may not guarantee an optimal solution. There's a possibility that following corrective labels confines the agent to a limited subset of expert-visited states, potentially leading to task failure.

However, we view our approach as a catalyst for promising future research:

- **Quantifying Local Continuity:** Further exploration into quantifying and capturing local continuity in the dynamics of robotic systems, especially through a data-driven approach.
- **Optimizing Dynamics Model Training:** The quality of the learned dynamics model directly affects the efficacy of generated labels, necessitating research into learning dynamics models while preserving continuity.
- **Handling Complex Discontinuities:** The application of CCIL in domains with intricate discontinuities, such as pixel-based images, remains an open area for investigation. Solutions may involve compact state representations and smoothness in latent spaces.
- **Real-world Robotics Application:** Extending our findings to real-world robotics presents an exciting frontier for future research, offering the potential for practical implementation and impact.

ACKNOWLEDGMENTS

We thank Sanjiban Choudhury for detailed feedback on the manuscript; Jack Umenberger for enlightening discussions; Khimya Khetarpal, Sidharth Talia and folks at the WEIRDLab at UW for edit suggestions. This work was (partially) funded by the National Science Foundation NRI (#2132848) and CHS (#2007011), the Office of Naval Research (#N00014-17-1-2617-P00004 and #2022-016-01 UW), and Amazon.

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

# A  GENERATING CORRECTIVE LABELS WITH KNOWN DYNAMICS FUNCTION

With a known dynamics function, we show an example algorithm that one can use to generate $N$ corrective labels (as defined in Sec 4.1). The algorithm first trains a behavior cloning policy, samples test time roll-out trajectories from the learned policy, and then derives labels using a root-finding solver.

---

**Algorithm 2** Generating Corrective Labels using Dynamics Function

---

1: **Input** Expert Data $\mathcal{D}* = (s_i^*, a_i^*, s_{i+1}^*)$.
2: **Input** Dynamics function $f(s^{i+1}|s^i, a^i)$.
3: **Input** Parameter $N$.
4: Initialize $\mathcal{D}^{\mathcal{G}} \leftarrow \varnothing, \mathcal{S}' \leftarrow \varnothing$
5: $\hat{\pi} = \arg\min_{\hat{\pi}} -\mathbb{E}_{s_i^*, a_i^*, s_{i+1}^* \sim \mathcal{D}^*} \log(\hat{\pi}(a_i^* \mid s_i^*))$
6: **for** $i \in 1..N$ **do**
7: $\quad s_0^{\mathcal{G}} \sim P_0, s_{j+1}^{\mathcal{G}} \sim f(s_j^{\mathcal{G}}, \pi(s_j^{\mathcal{G}})), \mathcal{S}' \leftarrow \mathcal{S}' \cup \{s_j^{\mathcal{G}}\}$
8: **end for**
9: **for** $i \in 1..N$ **do**
10: $\quad a_i^{\mathcal{G}} \leftarrow \arg\min_{a^{\mathcal{G}}} ||[s_i^{\mathcal{G}} + f(s_i^{\mathcal{G}}, a_i^{\mathcal{G}})] - s_k^*||$
11: $\quad \mathcal{D}^{\mathcal{G}} \leftarrow \mathcal{D}^{\mathcal{G}} \cup (s_i^{\mathcal{G}}, a_i^{\mathcal{G}})$
12: **end for**
13: **return** $\mathcal{D}^{\mathcal{G}}$

---

# B  PROOFS

## B.1  PROOF OF THEOREM 4.2

**Notation**. Let $f$ be the ground truth 1-step residual dynamics model, and let $\hat{f}$ be the learned approximation of $f$. Given a demonstration $(s_{t+1}^*, a_{t+1}^*)$, we have generated a corrective label $(s^{\mathcal{G}}, a_{t+1}^*)$.

**Assumptions**:

1. The estimation error of the learned dynamics model *at the training data* is bounded.
   $$\left\| f(s_{t+1}^*, a_{t+1}^*) - \hat{f}(s_{t+1}^*, a_{t+1}^*) \right\| \leq \epsilon.$$

2. $\hat{f}$ is locally $K_1$-Lipschitz in a neighborhood $\widehat{U}$ around $(s_{t+1}^*, a_{t+1}^*)$. i.e., for $s_t^{\mathcal{G}} \in \widehat{U}$:
   $$\left\| \hat{f}(s_t^{\mathcal{G}}, a_{t+1}^*) - \hat{f}(s_{t+1}^*, a_{t+1}^*) \right\| \leq K_1 \left\| s_t^{\mathcal{G}} - s_{t+1}^* \right\|$$

3. $f$ is locally $K_2$-Lipschitz in a neighborhood $U$ around $(s_{t+1}^*, a_{t+1}^*)$. i.e., for $s_t^{\mathcal{G}} \in U$:
   $$\left\| f(s_t^{\mathcal{G}}, a_{t+1}^*) - f(s_{t+1}^*, a_{t+1}^*) \right\| \leq K_2 \left\| s_t^{\mathcal{G}} - s_{t+1}^* \right\|$$

4. $s_t^{\mathcal{G}}$ is within the region of local continuity around $s_{t+1}^*$ for both $f$ and $\hat{f}$:
   $$s_t^{\mathcal{G}} \in U_{s_{t+1}^*} \cap \widehat{U}_{s_{t+1}^*}.$$

**Proof**:
$$\left\| f(s_t^{\mathcal{G}}, a_{t+1}^*) - \hat{f}(s_t^{\mathcal{G}}, a_{t+1}^*) \right\|$$
$$= \left\| f(s_t^{\mathcal{G}}, a_{t+1}^*) - f(s_{t+1}^*, a_{t+1}^*) + f(s_{t+1}^*, a_{t+1}^*) - \hat{f}(s_{t+1}^*, a_{t+1}^*) + \hat{f}(s_{t+1}^*, a_{t+1}^*) - \hat{f}(s_t^{\mathcal{G}}, a_{t+1}^*) \right\|$$
$$\leq \left\| f(s_t^{\mathcal{G}}, a_{t+1}^*) - f(s_{t+1}^*, a_{t+1}^*) \right\| + \left\| f(s_{t+1}^*, a_{t+1}^*) - \hat{f}(s_{t+1}^*, a_{t+1}^*) \right\| + \left\| \hat{f}(s_{t+1}^*, a_{t+1}^*) - \hat{f}(s_t^{\mathcal{G}}, a_{t+1}^*) \right\|$$
$$\leq \epsilon + (K_1 + K_2) \left\| s_t^{\mathcal{G}} - s_{t+1}^* \right\|$$

**Remark** Our assumption (1) about the error of the learned dynamics model is not a global constraint but simply requires the model to have a small prediction error *at the specific data point*. Our

assumptions (2) and (3) about the size of the local Lipschitz continuity is not a requirement on the global Lipschitz continuity in the dynamics, but simply requires space *near the expert support* to exhibit local continuity. Our proof leverages simple triangle inequality and is valid only near expert data support.

## B.2 PROOF OF THEOREM 4.3

**Notation**. Let $f$ be the ground truth 1-step residual dynamics model, and let $\hat{f}$ be the learned approximation of $f$. Given a demonstration $(s_t^*, a_t^*, s_{t+1}^*)$, we have generated a corrective label $(s^{\mathcal{G}}, a_t^* + \Delta, s_{t+1}^*)$.

**Assumptions**

1. $f$ is locally $K_S$-Lipschitz in *state* in a neighborhood $U_S$ around $(s_t^*, a_t^*)$.
   i.e., for $s_t^{\mathcal{G}} \in U_S$:
   $\left\| f(s_t^{\mathcal{G}}, a_t^*) - f(s_t^*, a_t^*) \right\| \leq K_S \left\| s_t^{\mathcal{G}} - s_t^* \right\|$.

2. $f$ is locally $K_A$-Lipschitz in *action* in a neighborhood $U_A$ around $(s_t^{\mathcal{G}}, a_t^*)$.
   i.e., for $a_t^* + \Delta \in U_A$:
   $\left\| f(s_t^{\mathcal{G}}, a_t^*) - f(s_t^{\mathcal{G}}, a_t^* + \Delta) \right\| \leq K_A \left\| \Delta \right\|$.

3. Using rejection sampling, we can enforce $\left\| s_t^{\mathcal{G}} - s_t^* \right\| \leq \epsilon_{rej}$.

4. Given that the generated labels come from a root solver:
$$s_{t+1}^* - \hat{f}(s_t^{\mathcal{G}}, a_t^* + \Delta) - s_t^{\mathcal{G}} = \epsilon_{opt} \text{ where } \epsilon_{opt} \to 0$$
$$s_t^* + f(s_t^*, a_t^*) - \hat{f}(s_t^{\mathcal{G}}, a_t^* + \Delta) - s_t^{\mathcal{G}} = \epsilon_{opt}$$
$$f(s_t^*, a_t^*) - \hat{f}(s_t^{\mathcal{G}}, a_t^* + \Delta) = s_t^{\mathcal{G}} - s_t^* + \epsilon_{opt}$$

5. The corrective label is within the neighborhood of local Lipschitz continuity of $f$:
   $s^{\mathcal{G}} \in U_S$ and $a_t^* + \Delta \in U_A$

**Proof**

$$\left\| f(s_t^{\mathcal{G}}, a_t^* + \Delta) - \hat{f}(s_t^{\mathcal{G}}, a_t^* + \Delta) \right\|$$
$$= \left\| f(s_t^{\mathcal{G}}, a_t^* + \Delta) - f(s_t^{\mathcal{G}}, a_t^*) + f(s_t^{\mathcal{G}}, a_t^*) - f(s_t^*, a_t^*) + f(s_t^*, a_t^*) - \hat{f}(s_t^{\mathcal{G}}, a_t^* + \Delta) \right\|$$
$$\leq \left\| f(s_t^{\mathcal{G}}, a_t^* + \Delta) - f(s_t^{\mathcal{G}}, a_t^*) \right\| + \left\| f(s_t^{\mathcal{G}}, a_t^*) - f(s_t^*, a_t^*) \right\| + \left\| f(s_t^*, a_t^*) - \hat{f}(s_t^{\mathcal{G}}, a_t^* + \Delta) \right\|$$
$$\leq K_A \left\| \Delta \right\| + K_S \left\| s_t^{\mathcal{G}} - s_t^* \right\| + \left\| s_t^{\mathcal{G}} - s_t^* \right\| + ||\epsilon_{opt}||$$
$$\leq K_A \left\| \Delta \right\| + (1 + K_S) \cdot \epsilon_{rej} + ||\epsilon_{opt}||$$

When the root solver yields a solution with $||\epsilon_{opt}|| = 0$, we have $\leq K_A \left\| \Delta \right\| + (1 + K_S) \cdot \epsilon_{rej}$

## C DETAILS FOR CCIL

Our proposed framework for generating corrective labels, **CCIL**, takes three steps:

1. **Learn a dynamics model**: fit a dynamics model $\hat{f}$ that is locally Lipschitz continuous.
2. **Generate labels**: solve a root-finding equation in Sec. 4.3 to generate labels.
3. **Augment the dataset and train a policy**: We use behavior cloning for simplicity to train a policy.

## C.1 LEARNING A DYNAMICS FUNCTION WHILE ENFORCING LOCAL LIPSCHITZ CONTINUITY

There are many function approximators to learn a model. For example, using Gaussian process can produce a smooth dynamics model but might have limited scalability when dealing with large amount

of data. In this paper we demonstrate examples of using a neural network to learn the dynamics model. There are multiple ways to enforce Lipschitz continuity on the learned dynamics function, with varying levels of strength that trade off theoretical guarantees and learning ability. In Sec. 4.2 we discuss a simple penalty to enforce local Lipschitz continuity, here we provide a few alternative methods, including the one used by Shi et al. (2019) and the one inspired by Gulrajani et al. (2017).

**Global Lipschitz Continuity via Spectral Normalization**. Using spectral norm with coefficient $L$ provides the strongest guarantee that the dynamic model is global $L$-Lipschitz. Concretely, spectral normalization Miyato et al. (2018) normalizes the weights of the neural network following each gradient update. Shi et al. (2019) used this method to train a dynamics function for drone navigation.

$$\arg\min_{\hat{f}} E_{s_j^*, a_j^*, s_{j+1}^* \sim \mathcal{D}^*} \left[ \text{MSE} \quad \text{while } W \to W/\max(\frac{||W||_2}{\lambda}, 1) \right] \tag{9}$$

However, spectral normalization enforces *global* Lipschitz bound. It may hinder the model's ability to learn the true dynamics.

**Local Lipschitz Continuity via Sampling-based Penalty**. Following Gulrajani et al. (2017), we can relax the global Lipschitz continuity constraint by penalizing any violation of the local Lipschitz constraint.

$$\arg\min_{\hat{f}} E_{s_j^*, a_j^*, s_{j+1}^* \sim \mathcal{D}^*} \left[ \text{MSE} + \lambda \cdot \mathbb{E}_{\Delta_s \sim \mathcal{N}} \max\left( \hat{f}'(s_j^* + \Delta_s, a_j^*) - L, 0 \right) \right]. \tag{10}$$

To estimate the local Lispchitz continuity, we can use a sampling-based method, $\mathbb{E}_{\Delta_s \sim \mathcal{N}} \max \left( \hat{f}'(s_j^* + \Delta_s, a_j^*) - L, 0 \right)$, which is indicative of whether the local continuity constraint is violated for a given state-action pair. The sampling-based penalty perturbs the data points by some sampled noise and enforces the Lipschitz constraint between the perturbed data and the original using a penalty term in the loss function.

Doing so ensures that the approximate model is mostly $L$ Lipschitz-bounded while being predictive of the transitions in the expert data. This approximate dynamics model can then be used to generate corrective labels.

**Local Lipschitz Continuity via Slack-variable**. We can allow for a small amount of discontinuity in the learned dynamics model in the same way that slack variables are modeled in optimization problems, e.g., SVM (Hearst et al., 1998) or max-margin planning with slack variables (Ratliff et al., 2006). With these insights in mind, we can reformulate the dynamics model learning objective as learning models that maximize likelihood (MSE) while minimizing the Lipschitz constant ($Lipschitz$).

$$\arg\min_{\hat{f}} \mathbb{E}_{s_j^*, a_j^*, s_{j+1}^* \sim \mathcal{D}^*} \text{MSE} + \lambda_j \cdot Lipschitz(s_j^*, a_j^*) + ||\lambda_j - \bar{\lambda}||_0$$
$$\text{where } ||\lambda_j - \bar{\lambda}||_0 \approx 1 - \exp\left( -\beta |\lambda_j - \bar{\lambda}| \right) \tag{11}$$
$$\text{and } Lipschitz(s_j^*, a_j^*) = \mathbb{E}_{\Delta_s \sim \mathcal{N}} \max\left( \hat{f}'(s_j^* + \Delta_s, a_j^*) - L, 0 \right).$$

To account for small amounts of discontinuity, we introduce a state-dependent variable $\lambda_j$ to allow violation of the Lipschitz continuity constraint. We minimize the number of non-zero entries in the slack variables (as noted by the L0 norm, $||\lambda_j - \bar{\lambda}||_0$) to ensure the model is otherwise as smooth as possible besides small amounts of local discontinuity. Building on Oliveira et al. (2021), we can use a practical, differentiable approximation for the L0 norm method, since the L0 norm itself is non-differentiable. Equipped with these locally continuous learned dynamics models, we can then generate corrective labels for robustifying imitation learning.

**Local Lipschitz Continuity via Weighted Loss**. Since we are optimizing a continuous function approximators, if the data contain subsets of discontinuity, such regions are likely to induce high training loss. We can create a slack variable for each data point, $\lambda_j$, to down-weight the on those regions. We then reformulate the dynamics model learning objective as learning models that maximize a weighted likelihood (MSE). $\sigma$ is the Sigmoid function:

$$\underset{\hat{f}}{\arg\min} \, \mathbb{E}_{s_j^*, a_j^*, s_{j+1}^* \sim \mathcal{D}^*} \, \sigma(\lambda_j) \cdot \text{MSE} + \sum \sigma(\lambda_j) \quad \text{while } W \to W/\max(\frac{||W||_2}{\lambda}, 1) \quad (12)$$

Intuitively, we expect that spectral normalization tends to work better for simpler environments where the ground truth dynamics are global Lipschitz with some reasonable $L$, whereas the soft constraint should be better suited for data regimes with more complicated dynamics and discontinuity. It remains a research question how best to train dynamics function for each domain and representation with different physical properties.

### C.2 GENERATING CORRECTIVE LABELS

In Sec. 4.3 we discuss two techniques to generate corrective labels. Depending on the structure of the application domain, one can choose to generate labels either by backtrack or by disturbed actions. Both techniques require solving root-finding equations (Eq. 5 and Eq. 7). To solve them, we can transform the objective to an optimization problem and apply gradient descent.

Eq. 5 specifies the root finding problem for backtrack labels.

$$s_t^* - \hat{f}(s_{t-1}^{\mathcal{G}}, a_t^*) - s_{t-1}^{\mathcal{G}} = 0.$$

Given $s_t^*, a_t^*$ and the learned dynamics function $\hat{f}$, we need to solve for $s_t^{\mathcal{G}}$ that satisfies the equation. We can instead optimize for

$$\underset{s_{t-1}^{\mathcal{G}}}{\arg\min} \, ||s_t^* - \hat{f}(s_{t-1}^{\mathcal{G}}, a_t^*) - s_{t-1}^{\mathcal{G}}|| \quad (13)$$

Similarly, we can transform Eq. 7 to become

$$\underset{s_t^{\mathcal{G}}}{\arg\min} \, ||s_t^{\mathcal{G}} + \hat{f}(s_t^{\mathcal{G}}, a_t^* + \Delta) - s_{t+1}^*|| \quad (14)$$

With access to the trained model $\hat{f}$ and its gradient $\frac{\partial \hat{f}}{\partial s_t^{\mathcal{G}}}$, one can use any optimizer. For simplicity, we use the Backward Euler solver that apply an iterative update $s_t^{\mathcal{G}} \leftarrow s_t^{\mathcal{G}} - s \cdot \frac{\partial \hat{f}}{\partial s_t^{\mathcal{G}}}$ where $s$ is a step size. We repeat the update until the objective is within a threshold $||s_t^{\mathcal{G}} + \hat{f}(s_t^{\mathcal{G}}, a_t^* + \Delta) - s_{t+1}^*|| \leq \epsilon_{opt}$.

### C.3 USING THE GENERATED LABELS

There are multiple ways to use the generated corrective labels. We can augment the dataset with the generated labels and treat them as if they are expert demonstrations. For example, for all experiments conducted in this paper, we train behavior cloning agents using the augmented dataset. Optionally, one can favor the original expert demonstrations by assigning higher weights to their training loss. We omit this step for simplicity in this paper.

Alternatively, one can query a trained imitation learning policy with the generated labels and measure the difference between our generated action and the policy output. This difference can be used as an alternative metric to evaluate the robustness of imitation learning agents when encountering a subset of out-of-distribution states. However, for a query state, our proposal does not necessarily recover *all* possible corrective actions. We defer exploring alternative ways of leveraging the generated labels to future work.

## D EXPERIMENTAL DETAILS

We provide details to reproduce our experiments, including environment specification, expert data, parameter tuning for our proposal and details about the baselines. We will also open-source the code and the configuration we use for each experiment, once the proposal is published.

## D.1 Environment and Task Design

We conduct experiments on 4 different domains and 8 robots, including the pendulum, a drone, a car, four robots for locomotion and one robot arm for manipulation. We consider 14 tasks: the pendulum, a modified pendulum swing task with discontinuity, three drone navigation tasks (fly-through, circle, hover), one LiDar racing task on F1tenth, four MuJoCo tasks (Hopper, HalfCheetah, Ant, Walker2D) and 4 MetaWorld tasks (coffee-pull, button-press-topdown, coffee-push, drawer-close). The F1tenth, MuJoCo and Metaworld environments are from open source implementations, and the drone environment is from a slightly modified open source implementation, discussed below. We will also describe how we set up the Pendulum environment and how we modify it to test our method with discontinuity.

**Pendulum Formulation.** The pendulum environment asks a policy to swing a pendulum up to the vertical position by applying torque. The properties of the system are controlled by the constants $g$, the gravitational acceleration, and $l$, the length of the pendulum. In all experiments, we take $g = 9.81$ and $l = 1$.

A pendulum's state is characterized by $\theta$, the current angle, and $\dot{\theta}$, the current angular velocity. To avoid any issues regarding angle representation, we do not directly store $\theta$ in the state representation; instead, we parameterize the state as $s = \begin{bmatrix} \sin\theta & \cos\theta & \dot{\theta} \end{bmatrix}^T$. A policy can control the system by applying torque to the pendulum, which we represent as a scalar $a$, which is clamped to the range $[-3, 3]$.

The continuous-time dynamics function is given by:

$$\frac{ds}{dt} = f(s, a) = \begin{bmatrix} \dot{\theta}\cos\theta \\ -\dot{\theta}\sin\theta \\ -\frac{g}{l}\sin\theta + a \end{bmatrix}.$$

This continuous dynamics model is then discretized to a timestep of 0.02 seconds using RK4. Additionally, although not required by the algorithms we study, we create the following reward function. where $\theta$ is the normalized pendulum angle in the range $[0, 2\pi)$ to metricize policy performance:

$$r(s, a) = -\frac{1}{2}\left\| \begin{bmatrix} \theta - \pi \\ \dot{\theta} \end{bmatrix} \right\|_2^2 - \frac{1}{2}a^2.$$

**Expert Formulation for Pendulum.** We formulate the expert policy using a combination of LQR and energy shaping control, where LQR is applied when the pendulum is near the top and energy-shaping is applied everywhere else. Note that the LQR gains were calculated by linearizing the dynamics around $\theta = \pi$, along with the cost function $c(s, a) = -r(s, a)$. So, the expert policy has the form:

$$\pi_e(s) = \begin{cases} -20.11(\theta - \pi) - 7.08\dot{\theta} & \text{if } |\theta - \pi| < 0.1 \\ -\dot{\theta}\left(\frac{1}{2}\dot{\theta}^2 - 9.81\cos\theta - 9.81\right) & \text{otherwise.} \end{cases}$$

**Discontinuous Pendulum.** We create a wall in the Pendulum environment to create local discontinuity. When the ball hits the wall, we *revert* the sign of its velocity, creating a discontinuity in the dynamics.

**Drone.** The drone environment used in our experiments is from a slightly modified fork of the original gym-pybullet-drones project. There are two notable modifications. Firstly, we added the circle task, where the agent aims to move in a circular trajectory. Second, we removed the floor, which better simulates a completely airborne drone and reduces training complexities caused by crashing into the floor.

## D.2 Demonstration Data

To feed expert data to train imitation learning agents, we design expert policies for the pendulum. For all other environments, we use the expert data from the D4RL dataset Fu et al. (2020). For drone

| Environment | Trajectories |
|---|---|
| Pendulum | 50 |
| Discontinuous Pendulum | 500 |
| F1tenth Racing | 1 |
| Drone hover | 5 |
| Drone circle | 30 |
| Drone fly-through | 50 |
| MuJoCo - Ant | 10 |
| MuJoCo - Walker2D | 20 |
| MuJoCo - Hopper | 25 |
| MuJoCo - HalfCheetah | 50 |
| MetaWorld, all tasks | 50 |

Table 4: Number of expert demonstration trajectories used in our experiments. We limit the amount of expert data to avoid making the task trivially solvable by naive behavior cloning.

environments, we first generate a bunch of via points alongside the target trajectories and then use a low-level PID controller to hit the via points one by one.

We note that it is possible to solve most tasks with naive behavior cloning if we feed them with a sufficient number of demonstrations. We thus limit the number of demonstrations we use for all tasks, as shown in Table. 4.

### D.3  PARAMETER TUNING.

Our proposal first trains a dynamics model. In this process, the most important hyper-parameter is $\bar{L}$, the enforced local Lipschitz smoothness per NN layer. Additional hyper-parameters depend on the exact protocol used to train the dynamics function and might include: $\lambda$ (weight of the Lipschitz penalty) and $\sigma$ (size of perturbation for estimating local Lipschitz continuity).

We first fit a dynamics model without enforcing any Lipschitz smoothness, to obtain an average prediction error for reference. To enforce local Lipschitz continuity, we then adopt the sampling-based penalty and train a series of dynamics models by sweeping parameters, $\bar{L} = [2, 3, 5, 10]$, soft dynamics $\lambda = [0.3, 0.5]$ and $\sigma = [0.0001, 0.0003, 0.0005]$.

To choose a dynamics model for generating labels, we will follow Theorem. 4.2 and Theorem. 4.3. We note that the local Lipschitz bound of a neural network is the product of the Lipschitz bound of each layer. Given that we are using two-layer NN to train our dynamics model, $L = \bar{L}^2$. We denote the empirical prediction error on the validation set for each trained dynamics model as $\epsilon$. We choose the best dynamics model that has the smallest error bound. For Theorem. 4.2, we optimize for $\epsilon + 2L \cdot ||C||$ where $C$ is a constant that we pick to be the average $s_{t+1}^* - s_t^*$ across the training data. For Theorem. 4.3, we optimize for $0.001 \cdot L + (1 + L)\epsilon$.

To generate corrective labels, we pick the dynamics model, the size of the perturbation (Sigma) and the rejection threshold (Epsilon). Empirically, Sigma = 0.00001 and Epsilon = 0.01 worked for all our environments. It is also possible to fine-tune Epsilon, the rejection threshold, for each task and each trained dynamics model to optimize the error bound. In our experiments, we omit this step for simplicity.

After generating corrective labels, we use two-layer MLPs (64,64) plus ReLu activation to train a Behavior Cloning agent with both original and augmented data.

### D.4  BASELINES

We evaluate the following algorithms to gain a thorough understanding of how our proposal compares to relevant baselines.

- EXPERT: For reference, we plot the theoretical upper bound of our performance, which is the score achieved by an expert during data collection.

- BC: a naive behavior cloning agent that minimizes the KL divergence on a given dataset.
- NOISEBC: a modification to naive behavior cloning that injects a small disturbance noise to the input state and reuses the action label, as described in Ke et al. (2021b).
- MOREL: Morel Kidambi et al. (2020) trains an ensemble of dynamics functions and uses the learned model for model-based reinforcement learning. It uses the variance between the model output as a proxy estimation of uncertainty to stay within high confidence region. We use the author implementation.
- MILO: MILO Chang et al. (2021) learns a dynamic function from given offline dataset in an attempt to mitigate covariate shift. We use the author's implementation in our experiment. MILO traditionally requires a large batch of offline dataset to train a dynamics function, we use only the expert demo dataset to train a dynamics function.
- CCIL: our proposal to generate high-quality corrective labels.

