# OpenReview forum: "CCIL: Continuity-Based Data Augmentation for Corrective Imitation Learning"
_ICLR.cc/2024/Conference — ICLR 2024 poster_

### Official Review · Reviewer_iEcJ · 2023-10-29

**Soundness:** 3 good
**Presentation:** 3 good
**Contribution:** 2 fair
**Rating:** 5
**Confidence:** 4

**Summary:**

The paper proposes to enhance the robustness of imitation learning methods by generating corrective data to account for compounding error and disturbances. Their work is based upon utilizing the local continuity in the environment dynamics. The paper augments the original expert's dataset with generated corrective labels within the neighborhood of the demonstrations but beyond the actual set of states and actions in the dataset. The authors' argue that this augmentation helps the agent to recover from perturbations and deal
with compounding error.

**Strengths:**

- Their problem is well-defined, and methods are explained clearly.
- Related work covers basic prior work.
- The data augmentation part where the authors utilize the local continuity of dynamics model helped them achieve better performance than the basic Behavioral Cloning Algorithm.

**Weaknesses:**

- While the writing was clear and easy to understand, the paper lacked substantial content. I didn't find any need to pause and think while reading and I skimmed through the paper rather quickly .
- Performance comparisons of their work are only done with basic Behavior Cloning and NoiseBC algorithms that are basic Imitation Learning (IL) Algorithms. Comparison with state-of the-art IL methods are missing.
-  I would recommend the authors to include experiments to compare the sample efficiency with other state of the art algorithms in terms of trajectories needed as that is also an important metric in IL paradigm.
- There are many Offline IL algorithms proposed recently in literature that have the same settings where they don't make any new interaction with the environment or the expert. Comparisons with them would be interesting to see.
- I would recommend the authors to also report results on Humanoid environment from Mujoco.

**Questions:**

Please check the Weaknesses part.

---

> ### Author Response · Authors · 2023-11-15
>
> Thank you for recognizing the novelty and soundness of our proposal. We address each of your concerns as follows:
>
> > “While the writing was clear and easy to understand, the paper lacked substantial content”
>
> Our proposal stems from a key observation - that the dynamics of robotic systems abides by physics laws and contains local continuities. While certainly simple, the literature in imitation learning has not investigated this form of structure for data augmentation and for alleviating the compounding errors. As seen from our empirical results, this data augmentation method is principled and makes a substantial difference in performance.
>
>
> > “Comparison with state-of-the-art IL methods are missing.”
>
> Per your request, we compared our method against two additional baselines: an imitation learning method [MILO](https://arxiv.org/abs/2106.03207) and model-based offline reinforcement learning method [MoREL](https://arxiv.org/abs/2005.05951). MILO is a state-of-the-art IL method which also learns a dynamic model from data and uses the learned model to learn from demonstrations. However, MILO usually demands access to a large dataset of transitions (in order to train a good dynamic model). In our experiments, we train MILO with only the given expert data. Similarly, model-based Offline RL requests additional access to a reward function as an assumption.
>
> Our results indicate that our method, CCIL, consistently outperforms these baselines in various tasks:
> - In the car domain, CCIL outperformed all other baselines.
> - In the drone domain (given the limited time, we used a subset of data for training), CCIL outperformed all other baselines.
> - In MuJoCo tasks, CCIL was the leading method on 3 out of 4 tasks.
> - In MetaWorld tasks, CCIL was competitive, tying with other baselines in 3 out of 4 tasks and slightly underperforming only in one task against MILO.
>
> **Mujoco**
> | | | | | |
> |--------------------------------------|:------------------------:|:------------------------:|:------------------------:|:------------------------:|
> |          | **Hopper** | **Walker**| **Ant** | **Halfcheetah**
> | CCIL | **3102.25 $\pm$309.25** | **4605.26 $\pm$ 129.02** |2073.60 $\pm$ 217.97 | **4182.15 $\pm$ 501.44**
> | VanillaBC | 2902.78 $\pm$689.64 | 3810.63 $\pm$ 828.23 |1646.24 $\pm$ 202.71 | 3872.82 $\pm$ 460.09
> | NoiseBC | 1563.56 $\pm$1012.02 | 2893.21 $\pm$ 1076.89 |**3160.51 $\pm$ 48.68** | 2044.24 $\pm$ 291.59
> | MOREL | 152.19 $\pm$34.12 | 70.27 $\pm$ 3.59 |1000.77 $\pm$ 15.21 | -2.24 $\pm$ 0.02
> | MILO | 566.98$\pm$100.32 | 526.72$\pm$127.99 |1006.53$\pm$160.43 |151.08$\pm$117.06
>
> **Metaworld**
> | | | | | |
> |--------------------------------------|:------------------------:|:------------------------:|:------------------------:|:------------------------:|
> |         | **CoffeePull** |**ButtonPress** | **CoffeePush**|**DrawerClose**
> | CCIL | **4168.46 $\pm$192.98** | **3775.22 $\pm$ 91.24** |**2484.19 $\pm$ 976.03** | 4145.45 $\pm$ 76.23
> | VanillaBC | 3552.59 $\pm$233.41 | **3693.02 $\pm$ 104.99** |1288.19 $\pm$ 746.37 | 2809.56 $\pm$ 439.70
> | NoiseBC | 3072.86 $\pm$785.91 | **3663.44 $\pm$ 63.10** | **2551.11 $\pm$ 857.79** | 4226.71 $\pm$ 18.90
> | MOREL |18.78 $\pm$0.09 | 14.85 $\pm$17.08 | 18.66 $\pm$ 0.02 |1222.2 3$\pm$ 1241.47
> | MILO | 232.49$\pm$110.44 | 986.46$\pm$105.79 | 230.62$\pm$19.37 | **4621.11$\pm$39.68**|
>
> **Car**
> | | | |
> |--------------------------------------|:------------------------:|:------------------------:|
> |    | **Succ.Rate** |**Avg. Score**
> | CCIL | **56.4%** | **0.75 $\pm$ 0.25**
> | VanillaBC | 31.9% | 0.58 $\pm$ 0.25
> | NoiseBC | 39.2% | 0.63 $\pm$ 0.27
> | MOREL | 0% | 0.001 $\pm$0.001
> | MILO | 0% | 0.21$\pm$0.003
>
> **Drone** (due to time constraint, we used ~50 trajectories for the drone tasks here)
> | | | | |
> |--------------------------------------|:------------------------:|:------------------------:|:------------------------:|
> |    | **Hover** |**Circle** |**FlyThrough**
> |CCIL  |-0.96E8 | -8.03E7 | -0.78E8
> |VanillaBC | -1.08E8 | -9.56E7 | -1.06E8
> |NoiseBC | -1.13E8 | -9.88E7 | -1.07E8
> |MOREL | -1.25E8 | -1.24E8 | -1.25E8
> |MILO | -1.26E8 | -1.25E8 | -1.25E8
>
>
>
> > “many Offline IL algorithms proposed recently in literature”
>
> Our proposal is meant to only leverage the given expert demonstration data, without access to reward or large amounts of offline data. Addressing your request, we just added [MILO](https://arxiv.org/abs/2106.03207) and [MoREL](https://arxiv.org/abs/2005.05951) for comparison. We would be happy to take a look if you could point us to other Offline IL algorithms with minimal assumptions.
>
>
> > “report results on the Humanoid environment”
>
> All our MuJoCo experiments used the expert data from the D4RL dataset which, unfortunately, did not provide demonstrations for the Humanoid. To keep the evaluations consistent and use an open-source dataset for MuJoCo, we did not include results on the Humanoid environment.

---

> ### Author Response · Authors · 2023-11-22
>
> Thank you for sharing valuable feedback to help improve our work. We have provided additional experiments and clarifications as requested. Please let us know whether our response has addressed your comments. We would be happy to engage in further discussions if needed.

---

> > ### Comment · Area_Chair_sdHN · 2023-12-05
> >
> > Reviewer iEcJ - please take a moment to read the final responses and decide if you would like to keep or change your rating. Thanks.

---

### Official Review · Reviewer_vWGV · 2023-10-30

**Soundness:** 3 good
**Presentation:** 4 excellent
**Contribution:** 3 good
**Rating:** 6
**Confidence:** 4

**Summary:**

This paper proposes the data augmentation method for behavioral cloning (BC) utilizing the local Lipschitz constraint. To train the forward dynamics from expert data, the proposed method (CCIL) minimizes mean-squared error with the regularization that is computed from the local Lipschitz constraint. Then, two techniques are proposed to generate transition triplets that can be used as expert data.  Once the dataset is augmented, naive BC is applied to find a policy. CCIL is evaluated on various tasks and outperforms BC and NoiseBC.

**Strengths:**

1. Although the proposed idea is simple, the experimental results show that CCIL is very powerful even if the environmental dynamics is not globally continuous.
2. The authors evaluated CCIL on various tasks, and it suggests that the proposed method is appealing to practitioners.
3. The manuscript is written well and easy to follow and understand.

**Weaknesses:**

1. My major concern is that the proposed method has to solve relatively complicated optimization problems. For example, Equation (3) contains two complicated terms: Lipschitz constraint and L0 norm. How to deal with the max operator in the Lipschitz constraint term is unclear.
2. The proposed method assumes a deterministic transition function. I am curious when the proposed method is applied to stochastic systems.

**Questions:**

1. The proposed method is formulated in a discrete-time state transition model, whereas the corresponding true system operates in continuous-time. Therefore, the proposed method implicitly applies a time discretization. In this case, the time interval is critical, and I think the Lipschitz constant depends on the time interval. How did the authors determine an appropriate Lipschitz constant? Or, are there any assumptions on the time discretization in the proposed method?
2. I do not fully understand the major differences between the techniques of the proposed method and Data as Demonstrator (DaD) proposed by Venkatraman et al. (2015)? I think the core idea is similar; therefore, it is worth discussing the advantages of the proposed method.
3. Two augmentation techniques are proposed, but I am unsure whether either would be equally useful. Is it possible to conduct an additional ablation study where one of the techniques is removed?
4. In the paragraph above Definition 4, the authors introduce $\mathrm{Support}(d^\pi)$, but it is not defined. Is $d^\pi$ a stationary distribution induced by $\pi$?
5. The first paragraph on page 5: $\hat{f}(s_t, a_t) \to s_{t+1}^* - s_t^*$ should be $\hat{f}(s_t^*, a_t^*) \to s_{t+1}^* - s_t^*$.
6. Is $\bar{\lambda}$ in Equation (3) is an average of $\{ \lambda_j \}_j$?
7. Please define $f'$ in Equation (3) explicitly.
8. Regarding the technique 1 (Backtrack label), what does "xlabel" mean?

---

> ### Author Response · Authors · 2023-11-17
>
> Thank you for acknowledging the strengths of our work, particularly the empirical results and the clarity of the manuscript. We address your questions as follows:
>
> > **W1: Complexity of Optimization Problems in the Proposed Method**
>
> Our approach requires learning a dynamic model while enforcing local Lipschitz continuity. We use a sampling-based estimation for the local Lipschitz continuity in Eq. 3. We implement the max term to penalize violations of Lipschitz continuity using the ReLU function. This method has empirically demonstrated an advantage over baselines without extensive hyper-parameter tuning. For tasks with largely local continuity, simpler techniques like Spectral Normalization can be used (details in Appendix C.1). Our method is flexible, allowing the use of any learned dynamics model that exhibits local continuity.
>
> > **Q1: How did the authors determine an appropriate Lipschitz constant?**
>
>
> Indeed the Lipschitz constant of the dynamics will depend on the time interval. Without assuming any prior knowledge about the system dynamics, we use empirical experiments to choose the desired local Lipschitz continuity for each task domain, as documented in Appendix D.3. Essentially, we tried constraining the Lipschitz continuity per each neural network layer to use different continuities (L=  2 / 3 / 5) and inspect the resulting training errors. Since the theoretical bound of our generated labels depends on both L and the training errors $\epsilon$ of the learned dynamics model (e.g., <= 0.001 · L + (1 + L)ϵ ), we choose the hyperparameter to yield the best theoretical bound.
>
> > **Q2: Comparison to Data as Demonstrator (DaD)**
>
> Similar to DaD, our method aims to generate “corrective” labels. However, our method is fundamentally different to DaD in (1) the assumptions and (2) the problem we are trying to solve.
>
> (1) We cited DaD in Section 2 paragraph 1 and explained that this method leveraged a form of **invariance** for data augmentation. Our method relies on the presence of local continuity instead. Our assumption is less strong and is applicable beyond position controlled domains.
>
> (2) DaD is a data augmentation method for training a dynamic model but not for policy. NoiseBC extends DaD to train a policy. It also leverages a form of known invariance (e.g., position control), injects noise disturbance and reuses the action label. Hence we include NoiseBC as a baseline, essentially reusing the ideas from DaD. We showed that our method could outperform NoiseBC in most domains we considered.
>
> > **Q3: Ablation Study on Augmentation Techniques**
>
> In our empirical evaluations we found that different tasks might benefit more from different techniques. For simplicity, in all our experiments, we used one of the techniques proposed. For example, on MuJoCo and MetaWorld tasks, NoisyAction technique outperforms Backward technique on half of the tasks. The benefits of using which technique to generate labels depends on the ground truth system dynamics and the learned dynamics function. We attach an additional ablation comparing the effectiveness of two techniques on Metaworld.
>
> **Metaworld**
> | | | | | |
> |--------------------------------------|:------------------------:|:------------------------:|:------------------------:|:------------------------:|
> |         | **CoffeePull** |**ButtonPress** | **CoffeePush**|**DrawerClose**
> | NoisyAction | **4168.46 $\pm$192.98** | 3758.89 $\pm$ 67.39  |**2484.19 $\pm$ 976.03** |  3988.67 $\pm$ 168.27
> | Backward | 3954.12$\pm$322.67 | **3775.22 $\pm$ 91.24** |1584.45$\pm$935.01 |**4145.45 $\pm$ 76.23**
> | VanillaBC | 3552.59 $\pm$233.41 | **3693.02 $\pm$ 104.99** |1288.19 $\pm$ 746.37 | 2809.56 $\pm$ 439.70

---

> > ### Author Response · Authors · 2023-11-17
> >
> > > **Q4: “Is $d^\pi$ a stationary distribution induced by $\pi$”**
> >
> > Yes, $d^\pi$ is the state distribution induced by the policy $\pi$ under the ground truth dynamics.
> >
> > > **Q5 “The first paragraph on page 5:  .. should be ..”**
> >
> > Thank you for pointing out the typo. We will correct it in the updated manuscript.
> >
> > > **Q6: “Is $\bar{\lambda}$ in EQ3 an average of lambda?”**
> >
> > $\bar{\lambda} is a chosen hyper-parameter.
> >
> > > **Q7: “Please define f’ in EQ3”**
> >
> > We apologize for the typo and confusion and will clarify the equation in the updated manuscript. In EQ3, we first estimates the local Lipschitz continuity around the expert data $(s_j^*, a_j^*)$ for the learned dynamic model $\hat{f}$ using a sampled noise $\delta_s$, such that  $\hat{f’}(s_j^*+\delta_s, a_j^*) \approx  \frac{ \hat{f}(s_j^*+\delta_s, a_j^*) - \hat{f}(s_j^*, a_j^*) } {|| \delta_s ||}$. We then record if the estimated Lipschitz continuity violates the given hyperparameter $L$ that enforces Lipschitz continuity:  $Lipschitz(s_j^*, a_j^*) = E_{\delta_s \sim \mathcal{N} } ReLU(\hat{f’}(s_j^*+\delta_s, a_j^*) - L) = E_{\delta_s \sim \mathcal{N} }  max(\hat{f’}(s_j^*+\delta_s, a_j^*) - L, 0)$
> >
> >
> > > **Q8: “Regarding technique 1 (Backtrack label), what does "xlabel" mean?”**
> >
> > Thank you for correcting our typo. "xlabel" should be "label".

---

> ### Author Response · Authors · 2023-11-22
>
> Thank you for sharing valuable feedback to help improve our work. We have provided additional experiments and clarifications as requested. Please let us know whether our response has addressed your comments. We would be happy to engage in further discussions if needed.

---

> > ### Comment · Area_Chair_sdHN · 2023-12-05
> >
> > Reviewer vWGV - please take a moment to read the final responses and decide if you would like to keep or change your rating. Thanks.

---

### Official Review · Reviewer_mpW3 · 2023-10-30

**Soundness:** 3 good
**Presentation:** 3 good
**Contribution:** 3 good
**Rating:** 6
**Confidence:** 4

**Summary:**

This study is dedicated to enhancing the robustness of imitation learning through the generation of corrective data, compensating for compounding errors and disturbances. Numerous experiments have been executed on an array of tasks, ranging from drone navigation and locomotion to robot manipulation, to validate the effectiveness of the proposed approach.

**Strengths:**

1. This work offers a detailed theoretical analysis, providing evidence that the quality of the generated label is bounded under specific assumptions related to the dynamics.

2. Various tasks ranging from drone navigation to locomotion and robot manipulation have been extensively experimented and analyzed in the study.

**Weaknesses:**

1. The proposed method is only compared to vanilla BC and noisy BC. The proposed method declares it constructs a dynamics model for policy learning and has used implementation with a model-based RL framework; therefore, it would be more robust to also include a comparison with other model-based RL methods. As model-based RL also constructs a dynamic model first before planning the most effective actions.

2. There is a lack of clarity in important implementation details. The process of generating corrective labels is discussed in Section 4 but the paper does not make it clear how these labels are employed in later stages. The additional corrective labels could be used to train the imitation learning agent, presumably a neural network? However, if this is the case, further details on the network's implementation could be discussed.

**Questions:**

The reason why noise BC underperforms compared to vanilla BC is not clear. If we reduce the noise added to the BC, noiseBC's performance should align more closely with that of vanilla BC. Nonetheless, in multiple tasks, noiseBC exhibits significantly poorer results. This could potentially be attributed to the fact that the added noise has not been not carefully chosen and thus, an excessive amount of noise has been injected into the system?

---

> ### Author Response · Authors · 2023-11-15
>
> Thank you for recognizing our theoretical contributions and comprehensive evaluations. To address your feedback, we have expanded our comparisons and clarified implementation details.
>
> **Clarification on Implementation Details**
>
> We augment the original training dataset with the newly generated labels. We then train imitation learning agents using these augmented data by feeding the data into a policy function. Our policy function predicts actions from input states. It is modeled by a neural network with 2 layers of 128 neurons each, using ReLU activation. Our implementation uses the BC algorithm from the open-source library [d3rlpy](https://takuseno.github.io/d3rlpy/).
>
> **Explanation of Noise BC's Underperformance**
>
> There are two main reasons why Noise BC underperforms compared to vanilla BC. First, certain domains may not be well-suited for NoiseBC. For instance, in drone navigation tasks using RPM control and in MuJoCo locomotion tasks that use torque control, reusing the action label after state perturbation may not be effective. Second, as you correctly pointed out, the level of noise introduced is crucial. Inappropriate noise levels can significantly impact performance.
>
> In our experiments, we experimented with the size of the noise (variance = 0.001, 0.0005, 0.0001). Our experiments currently applied a small noise level for NoiseBC across all tasks (noise variance = 0.0001). However, recognizing your concern, we experimented with a smaller noise level (variance = 0.00005) in two MuJoCo tasks (Hopper and Walker) where NoiseBC lagged behind vanilla BC. We observed that Noise BC's performance is indeed sensitive to the size of the noise. Reducing the noise slightly improved NoiseBC's performance, but it still did not reach the level of Vanilla BC. This suggests that the application of NoiseBC requires careful calibration of the noise level.
>
> These observations highlight the limitations of NoiseBC, further emphasizing the need for our proposed method.
>
> | | | |
> |--------------------------------------|:------------------------:|:------------------------:|
> |          | **Hopper** | **Walker**
> | NoiseBC (original, var=0.0001) | 1563.56 $\pm$1012.02 | 2893.21 $\pm$ 1076.89
> | NoiseBC (new, var=0.00005) | 2300.27 $\pm$823.85 | 2906.48 $\pm$ 326.78
> | VanillaBC | 2902.78 $\pm$689.64 | 3810.63 $\pm$ 828.23
>
> **Additional Baseline Comparisons**
>
> Per your request, we compared our method against two additional baselines: an imitation learning method [MILO](https://arxiv.org/abs/2106.03207) and model-based offline reinforcement learning method [MoREL](https://arxiv.org/abs/2005.05951).
>
> MILO is a state-of-the-art IL method which also learns a dynamic model from data and uses the learned model to learn from demonstrations. However, MILO usually demands access to a large dataset of transitions (in order to train a good dynamic model). In our experiments, we train MILO with only the given expert data. Similarly, model-based Offline RL requests additional access to a reward function as an assumption.
>
> Our results indicate that our method, CCIL, consistently outperforms these baselines in various tasks:
> - In the car domain, CCIL outperformed all other baselines.
> - In the drone domain (given the limited time, we used a subset of data for training), CCIL outperformed all other baselines.
> - In MuJoCo tasks, CCIL was the leading method on 3 out of 4 tasks.
> - In MetaWorld tasks, CCIL was competitive, tying with other baselines in 3 out of 4 tasks and slightly underperforming only in one task against MILO.

---

> ### Author Response · Authors · 2023-11-15
> **Additional Baseline**
>
> **Mujoco**
> | | | | | |
> |--------------------------------------|:------------------------:|:------------------------:|:------------------------:|:------------------------:|
> |          | **Hopper** | **Walker**| **Ant** | **Halfcheetah**
> | CCIL | **3102.25 $\pm$309.25** | **4605.26 $\pm$ 129.02** |2073.60 $\pm$ 217.97 | **4182.15 $\pm$ 501.44**
> | VanillaBC | 2902.78 $\pm$689.64 | 3810.63 $\pm$ 828.23 |1646.24 $\pm$ 202.71 | 3872.82 $\pm$ 460.09
> | NoiseBC | 1563.56 $\pm$1012.02 | 2893.21 $\pm$ 1076.89 |**3160.51 $\pm$ 48.68** | 2044.24 $\pm$ 291.59
> | MOREL | 152.19 $\pm$34.12 | 70.27 $\pm$ 3.59 |1000.77 $\pm$ 15.21 | -2.24 $\pm$ 0.02
> | MILO | 566.98$\pm$100.32 | 526.72$\pm$127.99 |1006.53$\pm$160.43 |151.08$\pm$117.06
>
> **Metaworld**
> | | | | | |
> |--------------------------------------|:------------------------:|:------------------------:|:------------------------:|:------------------------:|
> |         | **CoffeePull** |**ButtonPress** | **CoffeePush**|**DrawerClose**
> | CCIL | **4168.46 $\pm$192.98** | **3775.22 $\pm$ 91.24** |**2484.19 $\pm$ 976.03** | 4145.45 $\pm$ 76.23
> | VanillaBC | 3552.59 $\pm$233.41 | **3693.02 $\pm$ 104.99** |1288.19 $\pm$ 746.37 | 2809.56 $\pm$ 439.70
> | NoiseBC | 3072.86 $\pm$785.91 | **3663.44 $\pm$ 63.10** | **2551.11 $\pm$ 857.79** | 4226.71 $\pm$ 18.90
> | MOREL |18.78 $\pm$0.09 | 14.85 $\pm$17.08 | 18.66 $\pm$ 0.02 |1222.2 3$\pm$ 1241.47
> | MILO | 232.49$\pm$110.44 | 986.46$\pm$105.79 | 230.62$\pm$19.37 | **4621.11$\pm$39.68**|
>
> **Car**
> | | | |
> |--------------------------------------|:------------------------:|:------------------------:|
> |    | **Succ.Rate** |**Avg. Score**
> | CCIL | **56.4%** | **0.75 $\pm$ 0.25**
> | VanillaBC | 31.9% | 0.58 $\pm$ 0.25
> | NoiseBC | 39.2% | 0.63 $\pm$ 0.27
> | MOREL | 0% | 0.001 $\pm$0.001
> | MILO | 0% | 0.21$\pm$0.003
>
> **Drone**(due to time constraint, we used ~50 trajectories for the drone tasks here)
> | | | | |
> |--------------------------------------|:------------------------:|:------------------------:|:------------------------:|
> |    | **Hover** |**Circle** |**FlyThrough**
> |CCIL  |-0.96E8 | -8.03E7 | -0.78E8
> |VanillaBC | -1.08E8 | -9.56E7 | -1.06E8
> |NoiseBC | -1.13E8 | -9.88E7 | -1.07E8
> |MOREL | -1.25E8 | -1.24E8 | -1.25E8
> |MILO | -1.26E8 | -1.25E8 | -1.25E8

---

> ### Author Response · Authors · 2023-11-22
>
> Thank you for sharing valuable feedback to help improve our work. We have provided additional experiments and clarifications as requested. Please let us know whether our response has addressed your comments. We would be happy to engage in further discussions if needed.

---

> > ### Comment · Area_Chair_sdHN · 2023-12-05
> >
> > Reviewer mpW3 - please take a moment to read the final responses and decide if you would like to keep or change your rating. Thanks.

---

### Official Review · Reviewer_cJ4u · 2023-10-30

**Soundness:** 3 good
**Presentation:** 3 good
**Contribution:** 3 good
**Rating:** 6
**Confidence:** 3

**Summary:**

This work presents a method for augmenting imitation learning data by learning locally lipschitz-continuous dynamics models and then generating additional labels by perturbing the action to find noisy states as well as tracing states that would lead to the current state with the current action according to the learned dynamics model. Experiments on a diverse set of simulated tasks demonstrate the effectiveness of the proposed method.

**Strengths:**

Offline data augmentation is an important area of research that could lead to more robust  policies. This paper proposes an intuitive solution by generating additional data around existing data points by querying a locally smooth dynamics model.
The proposed solution categorizes two types of data augmentation: one by perturbing action labels and finding states that would land in the next state given this noise label, and the other by tracing states that would land in the current state given the current action.
This work presents thorough evaluation of the proposed method by experimenting with diverse task settings ranging from controlling a drone to manipulation tasks that have discontinuous dynamics.

**Weaknesses:**

The theoretical and algorithmic contribution is novel and exciting but the empirical results are not as impressive.

It would be great if the authors could test augmenting the training data with ground truth dynamics models (isn’t it deterministic -> computable given low-dimensional state representations?) to showcase the full potential of data-augmentation-based methods and situate the performance of the proposed method: i.e. help the audience understand if the performance gain/no-gain attribute to additional data or quality of the dynamics model.

This work could also benefit from additional experiments with varying number of demonstrations in a particular domain to show how much data is needed to learn a good dynamics model and at the same time could still benefit from additional augmentation data.

This work only conducted experiments in simulation, where dynamics models are deterministic and different from real applications. The authors should comment more on what challenges there would be to apply the proposed method in the real world and if one can benefit more or less from this paradigm of data augmentation.


----- Edit ------
The authors presented additional results during rebuttal that address some of my concerns about the evaluation. However, I do think a real-world experiment is practical and valuable for the true impact of this paper, given the proposed method is fully offline.

I am happy to raise my evaluation to weakly accept.

**Questions:**

See weakness for major concerns.

---

> ### Author Response · Authors · 2023-11-16
>
> Thank you for recognizing the novelty in our proposal and our thorough evaluations. Addressing your concerns, we have run more experiments to enhance the empirical evidence supporting our method, CCIL.
>
> **Ablation Experiment using Varying Number of Demonstrations**
>
> In response to your suggestion, we conducted an ablation study to vary the number of demonstrations on the Discontinuous Pendulum and the Drone environments.
>
> Our observations from both domains indicate that CCIL improves the performance of BC regardless of whether small or large amounts of data are used. This highlights a key advantage of our approach: CCIL does not require the learning of an extensive dynamics model using a large number of samples. As long as the learned model is well-fitted to the given demonstrations and the dynamic function exhibits local continuity around the data, CCIL can effectively generate corrective labels, thereby enhancing the robustness of imitation learning.
>
>
> *Discontinuous Pendulum*
> | | | | | |
> |--------------------------------------|:------------------------:|:------------------------:|:------------------------:|:------------------------:|
> |    | **500 demos** |**300 demos** |**100 demos** |**10 demos** |
> |CCIL  | -2556|-3194|-3275|-3807
> |VanillaBC  | -3204|-4075|-4213|-15236
>
>
> *Drone*
> | | | | |
> |--------------------------------------|:------------------------:|:------------------------:|:------------------------:|
> |    | **Hover** |**Circle** |**FlyThrough**
> |CCIL(~50 trajectories)  |-0.96E8 | -8.03E7 | -0.78E8
> |VanillaBC(~50 trajectories) | -1.08E8 | -9.56E7 | -1.06E8
> |CCIL(5000 traj) | -16458.96 | -2549.87 | -37060.3
> |VanillaBC(5000 traj) | -1.5E6 | -5.8E4 | -1.73E5
>
>
>
> **Ablation Experiment using the Ground Truth Dynamics**
>
> We conducted an ablation study to generate corrective labels using the ground truth dynamic model. This ablation requires that we have access to the ground truth dynamics, so we run it on the discontinuous Pendulum task. We run our experiment across 3 random seeds and found that on this simple domain, CCIL performs nearly as well as the ablation baseline that uses the ground truth dynamics:
> | | |
> |--------------------------------------|:------------------------:|
> |    | **Rewards** |
> |CCIL  | -2556
> |Ablation | -2453
>
>
>
>
> **Real-World Application Considerations**
>
> As we plan to apply our method to real robots, there are two key considerations.
>
> First, imitation learning policies in real-world scenarios are more susceptible to compounding errors due to sensor and actuator noise. Robustness would be a more critical issue to real world imitation learning. As highlighted by [1], the improvement brought by data augmentation techniques could be more prominent in real-world imitation learning policies than those observed in simulations.
>
> Second, real-world dynamics are often more complex than those in simulators due to contacts, friction cones, sensor and actuation noises. A significant challenge lies in how to effectively regularize the training of dynamics models for our approach. The successful application of CCIL in real-world settings will depend on our ability to accurately fit a dynamic model using real-world data.
>
> [1]: Ke, Liyiming, et al. "Grasping with chopsticks: Combating covariate shift in model-free imitation learning for fine manipulation." 2021 IEEE International Conference on Robotics and Automation (ICRA). IEEE, 2021.

---

> ### Author Response · Authors · 2023-11-16
>
> **Comparison with Additional Baselines:**
>
> To further highlight the effectiveness of CCIL, we conducted comparative experiments with two state-of-the-art methods:the [MILO](https://arxiv.org/abs/2106.03207) imitation learning method and the  [MoREL](https://arxiv.org/abs/2005.05951) model-based offline reinforcement learning method. Both these methods, like ours, construct a dynamic model from data for policy learning.  While MILO requires a large dataset of transitions, we trained it solely with the expert data available to us. MoREL, on the other hand, assumes additional access to a reward function. Our results indicate that our method, CCIL, consistently outperforms these baselines in various tasks.
>
> Our results indicate that our method, CCIL, consistently outperforms these baselines in various tasks:
> - In the car domain, CCIL outperformed all other baselines.
> - In the drone domain (given the limited time, we used a subset of data for training), CCIL outperformed all other baselines.
> - In MuJoCo tasks, CCIL was the leading method on 3 out of 4 tasks.
> - In MetaWorld tasks, CCIL was competitive, tying with other baselines in 3 out of 4 tasks and slightly underperforming only in one task against MILO.
>
> **Mujoco**
> | | | | | |
> |--------------------------------------|:------------------------:|:------------------------:|:------------------------:|:------------------------:|
> |          | **Hopper** | **Walker**| **Ant** | **Halfcheetah**
> | CCIL | **3102.25 $\pm$309.25** | **4605.26 $\pm$ 129.02** |2073.60 $\pm$ 217.97 | **4182.15 $\pm$ 501.44**
> | VanillaBC | 2902.78 $\pm$689.64 | 3810.63 $\pm$ 828.23 |1646.24 $\pm$ 202.71 | 3872.82 $\pm$ 460.09
> | NoiseBC | 1563.56 $\pm$1012.02 | 2893.21 $\pm$ 1076.89 |**3160.51 $\pm$ 48.68** | 2044.24 $\pm$ 291.59
> | MOREL | 152.19 $\pm$34.12 | 70.27 $\pm$ 3.59 |1000.77 $\pm$ 15.21 | -2.24 $\pm$ 0.02
> | MILO | 566.98$\pm$100.32 | 526.72$\pm$127.99 |1006.53$\pm$160.43 |151.08$\pm$117.06
>
> **Metaworld**
> | | | | | |
> |--------------------------------------|:------------------------:|:------------------------:|:------------------------:|:------------------------:|
> |         | **CoffeePull** |**ButtonPress** | **CoffeePush**|**DrawerClose**
> | CCIL | **4168.46 $\pm$192.98** | **3775.22 $\pm$ 91.24** |**2484.19 $\pm$ 976.03** | 4145.45 $\pm$ 76.23
> | VanillaBC | 3552.59 $\pm$233.41 | **3693.02 $\pm$ 104.99** |1288.19 $\pm$ 746.37 | 2809.56 $\pm$ 439.70
> | NoiseBC | 3072.86 $\pm$785.91 | **3663.44 $\pm$ 63.10** | **2551.11 $\pm$ 857.79** | 4226.71 $\pm$ 18.90
> | MOREL |18.78 $\pm$0.09 | 14.85 $\pm$17.08 | 18.66 $\pm$ 0.02 |1222.2 3$\pm$ 1241.47
> | MILO | 232.49$\pm$110.44 | 986.46$\pm$105.79 | 230.62$\pm$19.37 | **4621.11$\pm$39.68**|
>
> **Car**
> | | | |
> |--------------------------------------|:------------------------:|:------------------------:|
> |    | **Succ.Rate** |**Avg. Score**
> | CCIL | **56.4%** | **0.75 $\pm$ 0.25**
> | VanillaBC | 31.9% | 0.58 $\pm$ 0.25
> | NoiseBC | 39.2% | 0.63 $\pm$ 0.27
> | MOREL | 0% | 0.001 $\pm$0.001
> | MILO | 0% | 0.21$\pm$0.003
>
> **Drone** (due to time constraint, we used ~50 trajectories for the drone tasks here)
> | | | | |
> |--------------------------------------|:------------------------:|:------------------------:|:------------------------:|
> |    | **Hover** |**Circle** |**FlyThrough**
> |CCIL  |-0.96E8 | -8.03E7 | -0.78E8
> |VanillaBC | -1.08E8 | -9.56E7 | -1.06E8
> |NoiseBC | -1.13E8 | -9.88E7 | -1.07E8
> |MOREL | -1.25E8 | -1.24E8 | -1.25E8
> |MILO | -1.26E8 | -1.25E8 | -1.25E8
>
> We hope these additional experiments address your concerns. Let us know if you have any further question.

---

> ### Author Response · Authors · 2023-11-22
>
> **Preliminary Real Robot Result**
>
> We thank the reviewer for your feedback. To address your concern, we added a preliminary real world experiment, applying our proposal to a manipulator robot. On a peg insertion task that requires precision, CCIL achieved 40% success rate whereas BC had zero success rate, trained using 100 demonstrations. You can view an example rollout of CCIL at [imgur](https://imgur.com/a/TQANAJx)

---

> > ### Comment · Area_Chair_sdHN · 2023-12-05
> >
> > Reviewer cJ4u - please take a moment to read the final responses and decide if you would like to keep or change your rating. Thanks.

---

### Comment · Area_Chair_sdHN · 2023-11-20
**Please engage in reviewer-author discussions**

Reviewers - I encourage you to read the authors' response carefully and let the authors know whether their response has addressed your comments.

---

### Author Response · Authors · 2023-11-23
**Summary of Reviewer Feedback and Author Responses**

We thank all reviewers for their valuable feedback.

Reviewers acknowledged the detailed theoretical analysis and novelty of our method (mpW3 and cJ4u), highlighted the clarity of our presentation (vWGV and iEcJ), praised the extensive experimentation and analysis (mpW3 and vWGV) and recognized the empirical performance of our approach to improve imitation learning agents (iEcJ).

To address reviewers' concerns, we conducted additional experiments (1) with two new baselines from imitation learning and offline reinforcement learning, (2) using varying numbers of demonstrations and (3) using ground truth dynamics models. We added a significant number of experiments to demonstrate our method's empirical effectiveness in varying domains and conditions. For real-world applicability, we added a preliminary experiment on a manipulator robot, showing promising results. We provide clarifications on the implementation details of our policy function and the Noise BC's underperformance.

We have made significant efforts to address all raised concerns and questions. However, it's noteworthy that only one out of four reviewers responded to our rebuttal and adjusted their scores. We respectfully request further engagement from the reviewers, as we believe our additional experiments and clarifications strongly support the novelty, effectiveness and potential impact of our work.

---

### Meta-Review · Area_Chair_sdHN · 2023-12-06

**Metareview:**

This paper proposes a simple method for augmenting imitation learning data by assuming that the environment dynamics are locally Lipschitz-continuous. This approach aims to enhance the robustness of imitation learning by generating corrective action labels that compensates for compounding errors and disturbances. This is achieved by learning locally Lipschitz-continuous dynamics models and then generating additional labels through action perturbations and state tracing. The proposed method is evaluated against drone navigation, driving, robot locomotion, and manipulation. In general, the reviewers appreciate the writing clarity and good empirical results in more continuous environments despite a simple idea. After rebuttal, some minor weaknesses have not been addressed:
* As the authors target real-world applications, it's important to consider the suggestion of including experiments to compare the sample efficiency with other state of the art algorithms in terms of trajectories needed (Reviewer iEcJ). This remains unanswered.
* The locomotion and manipulation results are not impressive. Although that's expected because of dynamics incontinuity and explained in the paper, it could be emphasized more, even as a negative result, in future revision.

The AC considers the merits outweighing the flaws and thus recommends accepting this paper.

**Justification For Why Not Higher Score:**

Lipschitz-continuity is a strong assumption, which limits the applicability and impact of this work.

**Justification For Why Not Lower Score:**

This paper presents a simple yet effective approach

---

### Decision · Program_Chairs · 2024-01-16

Accept (poster)